# Theory-Grounded Evaluation of Human-Like Fallacy Patterns in LLM Reasoning

**Andrew Keenan Richardson**[1], **Ryan Othniel Kearns**[1,2,†], **Sean Moss**[1,3],
**Vincent Wang-Maścianica**[1], **Philipp Koralus**[1]
[1]The Laboratory for Human-Centered AI, Institute for Ethics in AI,
  Faculty of Philosophy, University of Oxford, UK
[2]Oxford Internet Institute, University of Oxford, UK
[3]School of Computer Science, University of Birmingham, UK

## Abstract

We study logical reasoning in language models by asking whether their errors follow established human fallacy patterns. Using the Erotetic Theory of Reasoning (ETR) and its open-source implementation, PyETR, we programmatically generate 383 formally specified reasoning problems and evaluate 38 models. For each response, we judge logical correctness and, when incorrect, whether it matches an ETR-predicted fallacy. Two results stand out: (i) as a capability proxy (Chatbot Arena Elo) increases, a larger share of a model's incorrect answers are ETR-predicted fallacies ($\rho = 0.360, p = 0.0265$), while overall correctness on this dataset shows no correlation with capability; (ii) reversing premise order significantly reduces fallacy production for many models, mirroring human order effects. Methodologically, PyETR provides an open-source pipeline for unbounded, synthetic, contamination-resistant reasoning tests linked to a cognitive theory, enabling analyses that focus on error composition rather than error rate.

## 1 Introduction

Language models increasingly solve complex tasks (Wei et al., 2023; Kojima et al., 2022). We ask a simple question: when they err on controlled reasoning problems, do their errors align with human fallacies? Human reasoning shows systematic, repeatable fallacies (Tversky & Kahneman, 1974; Kahneman & Tversky, 1982; Evans, 1994; Walsh & Johnson-Laird, 2004; Johnson-Laird, 2006). These are not random mistakes; they follow predictable patterns across tasks. Understanding whether LLM errors share these patterns is useful both scientifically and for deployment in settings that require reliable reasoning (Jacobs, 2021; Bommasani et al., 2022).

To rigorously investigate this question, we leverage the Erotetic Theory of Reasoning (ETR) (Koralus, 2022; Mascarenhas & Koralus, 2017a; Koralus & Mascarenhas, 2013), a formal cognitive theory that precisely predicts human reasoning patterns across multiple domains. ETR posits that human reasoning operates by maintaining disjunctive alternatives and filtering these alternatives when taking on new information, a process that can systematically lead to error. ETR provides formal specifications of both how and when humans will make specific reasoning errors, allowing us to generate arbitrary reasoning problems with predictable failure patterns.

ETR has an open-source implementation, PyETR (Koralus et al., 2025). Using PyETR, we mechanically generate reasoning problems predicted to elicit specific fallacies. We evaluate 38 models on a fixed corpus of 383 such problems.

Our experimental investigation evaluates 38 language models ranging from smaller models (e.g., `Mistral 7B Instruct v0.1`) to larger systems (e.g., `GPT-4.5`, `Claude 3.7`), treating Chatbot Arena Elo as one capability proxy (Chiang et al., 2024). We observe that as this proxy increases, a larger share of logically incorrect answers align with ETR-predicted human fallacies ($\rho = 0.360, p = 0.0265$). This is a correlative claim; our experiments do not suggest or support any causal mechanism.

[†]ryan.kearns@oii.ox.ac.uk

While language models often improve on many benchmarks (Wei et al., 2023; Bubeck et al., 2023), our results indicate that within our task domain, the proportion of errors matching ETR-predicted fallacies increases with Chatbot Arena Elo as a capability proxy. These findings suggest an overlap between LLM error patterns and human fallacy patterns irrespective of any underlying cognitive mechanisms.

We affirm the robustness of our result using metrics other than Chatbot Arena Elo, where available, for a subset of the 38 models. We find significant Spearman correlation with estimates of training compute (Epoch AI, 2025) ($\rho = 0.489, p = 0.0334$) and significant exponential fit for mean score on the HELM capabilities benchmark (Liang et al., 2023) ($r = 0.796, p = 0.0103$).

**Contributions** We report a statistically significant correlation between Chatbot Arena Elo as a capability proxy and reasoning errors predicted by ETR, an empirically grounded formal theory of human reasoning. We also provide the first substantial application of PyETR—an open-source implementation of ETR—to generate unbounded, synthetic, contamination-resistant reasoning tasks linking model errors to theory-predicted fallacies.

The remainder of the paper is organised as follows. Section 2 introduces ETR and PyETR. Section 3 details methodology. Section 4 presents results. Section 5 discusses implications and concludes.

**Related Work** Substantial prior literature in psychology and cognitive science documents irrationalities in human judgement and decision-making (Tversky & Kahneman, 1974; Kahneman & Tversky, 1982; Evans, 1994; Kahneman et al., 1990). In particular, the psychology of deductive inference reveals human propositional reasoning to be vulnerable to logically-irrelevant linguistic effects, like co-references between premises (Walsh & Johnson-Laird, 2004). Mental model theory provides an explanatory paradigm for these results (Johnson-Laird & Byrne, 1991), though prior work on the Erotetic Theory of Reasoning matches and has replicated mental model predictions on deductive inference tasks (Mascarenhas & Koralus, 2015; 2017b).

In the language model literature, prior studies find that while modern LLMs largely succeed at syllogistic inference (Clark et al., 2021; Eisape et al., 2024; Bertolazzi et al., 2024), they exhibit human-like reasoning failures, including distractability (Shi et al., 2023), content effects (Lampinen et al., 2024), and order effects (Eisape et al., 2024; Saparov & He, 2023). In a prior work by two of the present authors, the first application of the Erotetic Theory to LLMs found the predictive utility of the theory to increase for larger model sizes (Koralus & Wang-Maścianica, 2023), though this study was limited to OpenAI's GPT model family (Brown et al., 2020; OpenAI, 2022; 2024) and did not make use of PyETR. Our implementation yields an endlessly regenerative and content-agnostic test of the Erotetic Theory for LLMs in a manner more resilient to data contamination.

## 2 THE EROTETIC THEORY OF REASON

Modern accounts of human reasoning highlight a duality: impressive competence and predictable errors. The Erotetic Theory of Reasoning (ETR) explains both with a single idea: we reason by managing questions and candidate answers (or "alternatives") and filtering them when new information arrives. Filtering is efficient but can drop relevant alternatives too early, producing characteristic fallacies. ETR formalises this process and defines when conclusions stabilise against follow-up questions (a state called "erotetic equilibrium").

At a high level,[1] ETR considers reasoning to consist of three steps: (1) maintain disjunctive alternatives as candidate answers to an implicit question, (2) filter by best match with incoming information (risking premature elimination), and (3) recover by asking questions (raising structured alternatives) that reintroduce necessary information. This theory predicts a range of empirical reasoning data across propositional and first-order logic, probability, and decision-making. Morover, it is mathematically proved in Koralus (2022) that ETR converges to normative standards of rationality in the limit, as the recovery step introduces enough alternatives to stabilise further reasoning steps.

We can demonstrate ETR with a compact example (Example 49 in Koralus (2022)).

---

[1]A full formalism for ETR is provided in Appendix A.

| **Premise 1** | Either there is an ace in Mary's hand and some other player has a king, or else there is a queen in John's hand and some other player has a jack. |
|---|---|
| **Premise 2** | Sally has a king. |
| **Question** | Does it follow that Mary has an ace? |

The correct logical answer is "not necessarily", yet many respondents endorse "Mary has an ace" (60% in data following Mascarenhas & Koralus (2017a)).

To model the problem in logic, we would first choose a representation of the basic predicates, in this case $\mathsf{Ace}(x)$ for the predicate that (the card) $x$ is an ace, and $\mathsf{Has}(y, z)$ for the predicate that (player) $y$ holds (a card) $z$. The same applies in ETR, where instead of logical formulas we have the central concept of a *view* (see Appendix A) for the full mathematical formalism). It will suffice for now to use the (still precise) shorthand notation to express the premises and conclusion as ETR views:[2]

| **P1** | $\{\mathsf{Ace}(a^*)\mathsf{Has}(\mathsf{Mary}, a)\mathsf{King}(b^*)\mathsf{Has}(c, b), \mathsf{Queen}(d)\mathsf{Has}(\mathsf{John}, d)\mathsf{Jack}(e)\mathsf{Has}(f, e)\}$ |
|---|---|
| **P2** | $\{\mathsf{King}(g^*)\mathsf{Has}(\mathsf{Sally}, g)\}$ |
| **Q** | $\{\mathsf{Ace}(h^*)\mathsf{Has}(\mathsf{Mary}, h)\}$ |

Each view is given as a disjunctive set of alternatives, separated by a comma. Each alternative is to be considered as a conjunctive set of atomic propositions, where for convenience we omit the set-brackets and commas and simply juxtapose elements. Many readers will recognise these as *disjunctive normal forms*, but in ETR there is no inherent ordering to the atoms in a conjunction or alternatives in the disjunction. The individual lowercase letters can be understood as existentially quantified variables, with the scope of the quantification encompassing the entire disjunction ("prenex form"). A view may also carry a supposition, written as a superscript (see Table 1 for examples). Note that since any formula of first-order logic can be written in prenex disjunctive normal form, ETR is as expressive as full first-order logic.

PyETR, an open-source software package that serves as a calculator for ETR (Koralus et al., 2025), can parse views from a similar text format.

```
from pyetr import View
p1 = View.from_str("∃a ∃b ∃c ∃d ∃e ∃f {Ace(a*)Has(Mary(),a)Has(c,b
    )King(b*),Has(John(),d)Has(f,e)Jack(e)Queen(d)}")
p2 = View.from_str("∃g {King(g*)Has(Sally(),g)}")
q = View.from_str("∃h {Ace(h*)Has(Mary(),h)}")
```

Note the requirement to explicitly declare existentially quantified names (the letter E may be used in place of ∃). ETR posits some default procedures for ordinary reasoning built out of basic operations. The workhorse among the basic operations is Update, which updates a current view with an incoming one. We can check in PyETR that ETR does predict a fallacy here, because Update discards the seemingly less relevant alternative where John has a queen.

```
>>> v = p1.update(p2); print(v)
∃g ∃l ∃m {Ace(l*)Has(Mary(),l)Has(Sally(),g)Has(m,g)King(g*)}
>>> v.query(q)
∃h {Ace(h*)Has(Mary(),h)}
```

The actual default procedure of ETR involves a few more steps which become relevant in more complex problems. The full version is implemented in PyETR.

```
>>> from pyetr.inference import default_procedure_does_it_follow
>>> default_procedure_does_it_follow([p1,p2],q)
True
```

---

[2]An additional aspect not present in ordinary logic is *issue structure*. This is notated with an asterisk on some occurrences of terms, indicative of being 'at issue' for the containing atomic proposition. The issue structure has no logical content, but plays a role in guiding ETR inferences. Determination of issue structure from cues in natural language or context is outside the scope of ETR, but a simple heuristic in this example is that the repetition of 'Ace' and 'King' suggests those concepts are at issue.

## 3 METHODOLOGY

In our experiment, a *reasoning problem* is given by a list of views (the *premises*), and the problem is to answer the question "what, if anything, follows?". Our method generates reasoning problems where the ETR-predicted answer is a logical fallacy.

### 3.1 DATA PROCESSING

Table 1: Original bank of reasoning problems.

| Problem | Symbolic representation | Natural language example |
|---|---|---|
| Modus ponens | $\{R(x)\}^{\{Q(x)\}}$ 
 $\{Q(x)\}$ 
 $\therefore \{R(x)\}$ | If it rains, it's wet. 
 It rains. 
 Therefore, it's wet. |
| Modus tollens | $\{R(x)\}^{\{Q(x)\}}$ 
 $\{\overline{R(x)}\}$ 
 $\therefore \{\overline{Q(x)}\}$ | If the switch is on, then the lamp is lit. 
 The lamp is not lit. 
 Therefore, the switch is not on. |
| Quantified modus ponens | $\forall x \{R(x)\}^{\{Q(x)\}}$ 
 $\forall x \{Q(x)\}^{\{P(x)\}}$ 
 $\therefore \forall x \{R(x)\}^{\{P(x)\}}$ | All mammals have lungs. 
 All dogs are mammals. 
 Therefore, all dogs have lungs. |
| Disjunction fallacy | $\{Q(x)R(x), S(x)T(x)\}$ 
 $\{Q(x)\}$ 
 $\therefore \{R(x)\}$ | She is quiet and clever, or tall and athletic. 
 She is quiet. 
 Therefore, she is clever. |

We source our original bank of reasoning problems from examples presented in the *Reason and Inquiry* text (Koralus, 2022). These include basic "template" reasoning problems like modus ponens, modus tollens, and the disjunction fallacy as originally presented in Walsh & Johnson-Laird (2004). Table 1 presents the complete list of original problems. Not all reasoning problems take the form of valid syllogistic inferences; that is, not all conclusions from ETR inference deductively follow from the premises.

To avoid complications in the mapping to natural language, we restrict to views with only *monadic* (one-place) predicates. The corresponding monadic first-order logic is known to be less expressive than full first-order logic, but still allows for a rich set of reasoning problems. Our method could easily be extended to include polyadic predicates.

### 3.2 GENERATIVE PIPELINE

We define a collection of mutation functions that give slight modifications to ETR view objects. These functions serve to expand our bank of reasoning problems by modifying existing problems by single premises. Table 2 presents the complete list of possible mutation rules.

Generating a new problem proceeds by iteratively building a list of views (the premises in the problem). A new view is selected at random from the original problem bank and subjected to a random number of random mutation rules. The mutated view is added to the list if ETR predicts the problem admits a non-trivial answer. New views are added until all of the following stopping conditions are met: (1) The problem is the right size, meaning that summing the number of atoms in each view gives between 4 and 11. If 11 is exceeded, backtracking is used. (2) The ETR-predicted conclusion contains a single categorical alternative (without disjunctions). (3) The ETR-predicted conclusion is a logical fallacy. To mitigate concerns about dataset-specific effects, we intentionally generated problems across diverse domains and structures. The PyETR-based generation ensures our problems test reasoning capacities rather than memorization of training examples.

**Dataset curation** We initially generated 400 problems. Pre-analysis integrity checks identified 17 items whose fallacy status did not meet the above stopping conditions, so we excluded them prior

Table 2: Mutation rules for ETR views.

| Mutation Type | Description |
|---|---|
| Predicate Addition | Introduces a new predicate symbol to the language. |
| Constant Addition | Introduces a new constant symbol to the formal system. |
| Variable Addition | Adds a new variable to the set of arbitrary objects. |
| Constant-to-Variable Substitution | Replaces constants with $\forall$ or $\exists$ quantified variables. |
| Conjunctive Atom Insertion | Conjoins new atomic formulas to existing states. |
| Disjunctive State Addition | Disjunctively creates new states with atoms. |
| Atom Negation | Applies or removes negation from atomic formulas. |

to all analyses, yielding 383 problems. Given the modest reduction in sample size and the cost of re-running all model evaluations, we report results on this vetted 383-set; the generation pipeline supports regenerating larger sets for future work.

## 3.3 MAPPING TO NATURAL LANGUAGE

To evaluate logical reasoning in language models in a more naturalistic context, we developed a systematic approach to convert formal logical statements into natural language prompts. We created 12 themes for natural language framings of logical problems, such as a researcher figuring out the properties of a novel element or a newly discovered creature. The elements of our 12 themes are listed in Appendix B. For each theme, we established consistent mappings between logical elements and thematic equivalents. The thematic conversion process serves multiple purposes: it ensures our results are robust against content effects (Lampinen et al., 2024), it enables us to test whether the reasoning patterns are consistent in diverse contexts, and it mitigates potential data contamination, a significant problem for LLM evaluation (Zhou et al., 2023; Deng et al., 2024; Li & Flanigan, 2024; Singh et al., 2024) by reformulating problems in novel scenarios unlikely to appear in training data.

Consider the following example of a disjunction fallacy problem in its formal representation:

| **P1** | $\{Q(x)R(y), S(x)T(y)\}$ |
|---|---|
| **P2** | $\{Q(x)\}$ |

In the "alchemy" theme, logical predicates (e.g., $Q(x)$, $R(y)$) were mapped to thematic attributes (e.g., "is transmuting", "is time-bending"), while logical variables (e.g., $x$, $y$) were mapped to imaginary thematic entities (e.g., "cosmic dust", "vital mercury"). Using our alchemy theme, this was transformed into:

> I'm an alchemist studying mysterious substances in my laboratory. I need to understand their properties through logical reasoning. Here's what I've discovered:
>
> - Cosmic dust is transmuting and vital mercury is time-bending, or cosmic dust is immortality-granting and vital mercury is spirit-affecting.
> - Also, we know that cosmic dust is immortality-granting.
>
> What if anything follows? I do not have an intended answer in mind, and it is possible that nothing follows. Please be succinct and precise.

This is typical of the prompts sent to the LLM.[3] It includes a thematic preamble and a natural language prompt to reason in an open-ended fashion with: "What if anything follows?" Each problem maintained the same prompt structure: an introduction establishing the thematic context, followed by the premises expressed in natural language, and concluding with a standardised question asking

---

[3]Complete prompts, with reasoning traces and responses, are provided in Appendix D.

what follows from the premises. The prompt explicitly stated that nothing might follow, to avoid biasing models toward making unwarranted inferences (Saparov & He, 2023).

## 3.4 DATA COLLECTION

**Model Selection**  We evaluated 38 language models of varying sophistication, including several specifically designed as reasoning models. Selection informally prioritized quantity, popularity of usage, and representation of a diverse range of scores. Chatbot Arena scores were obtained from `https://huggingface.co/spaces/lmarena-ai/chatbot-arena-leaderboard/resolve/main/elo_results_20250423.pkl`. Scores for the HELM Capabilities benchmark for 9 models were obtained from `https://crfm.stanford.edu/helm/capabilities/v1.10.0/#/leaderboard` (Liang et al., 2023) and estimates for training compute for 19 models were obtained from `https://epoch.ai/data/ai-models` (Epoch AI, 2025) (both URLs accessed 30 July 2025).

**Response Processing and Filtering**  Models provided conclusions in natural language, which we converted to PyETR format for evaluation. To ensure fair assessment of reasoning (not formatting), GPT-4.1-mini served as our translation layer without access to the original premises. Our manual spot-checks confirmed this translation step to be perfectly faithful.

Each model was evaluated on 383 reasoning tasks generated by PyETR. During analysis, we encountered occasional parse errors when processing model responses. To maintain data quality while preserving statistical power, models with parse error rates exceeding 20% across the test set were excluded from analysis, which resulted in the exclusion of one model (`google_gemini-2.5-pro-preview-03-25`). For the remaining models, individual responses that produced parse errors were excluded from analysis on a case-by-case basis rather than discarding all responses from that model.

**Testing Framework**  The full set of logical problems were framed as evals, using Eleuther's Language Model Evaluation Harness (Gao et al., 2024), which handles evaling and metric collection against a wide range of models. OpenRouter (OpenRouter, 2025) was used for all models in order to have a unified API. This consumed less than $1,000 of compute resources. Each model was given a token limit of 3,000 output tokens.

To ensure a fair comparison across architectural differences, all reasoning models were allocated 2,400 thinking tokens during evaluation. This standardization was justified given our use of the Chatbot Arena Elo score as our primary capability metric, which already accounts for performance differences regardless of underlying mechanisms (Chiang et al., 2024).

**Key Measures**  We collected the following measures to evaluate both the logical correctness of model responses and their alignment with human-like reasoning patterns, enabling our analysis of how these patterns correlate with model sophistication. A model's answer was considered **'logically correct'** if it was a logical consequence of the premises. This was tested using PySMT (Cimatti et al., 2017) to check whether the negation of the conclusion was inconsistent with the premises. LLM responses were classified as **'ETR-predicted'** if ETR predicts an endorsement of that conclusion according to PyETR's implementation of `default_procedure_does_it_follow`.

**Statistical Tests**  We selected Pearson correlation to capture linear relationships between variables while also reporting Spearman's rank correlation to account for potentially non-linear monotonic relationships without assuming normality in the distribution of fallacy rates. For our premise-order intervention analysis (discussed in Section 4.4), we employed two-proportion z-tests to rigorously evaluate whether reversing premise order produced statistically significant changes in fallacy rates. The z-test was specifically chosen because it allows us to determine if the observed differences in proportions (original vs. reversed premise order) represent genuine effects rather than random variation, providing a reliable measure of the significance of order effects across different model capabilities. This statistical approach enables us to quantify the extent to which models exhibit the same premise-order sensitivity documented in human reasoning studies.

## 4 EXPERIMENTAL RESULTS

Our analysis reveals several notable patterns in how language models exhibit human-like reasoning fallacies. We report the results of Pearson correlation tests, as well as Spearman tests for the detection of non-parametric monotonic relationships. We consider $p \leq 0.05$ a suitable boundary for statistical significance.

### 4.1 PROPORTION OF ERRORS THAT ARE FALLACIES

We operationalise a "human-like fallacy" (or simply "fallacy") as a response that is both ETR-predicted and logically incorrect. This definition captures instances where models make the same systematic reasoning errors that ETR predicts humans would make. Formally, for a given model $m$ and reasoning problem $p$, we define:

$$\text{HumanLikeFallacy}(m, p) = \begin{cases} 1 & \text{if ETR-predicted}(m, p) \wedge \neg \text{LogicallyCorrect}(m, p) \\ 0 & \text{otherwise} \end{cases}$$

Our metric of interest is the *fallacy rate*, defined as the proportion of human-like fallacies to total logically incorrect answers for each model:

$$\text{FallacyRate}(m) = \frac{\sum_{p \in P} \text{HumanLikeFallacy}(m, p)}{\sum_{p \in P} \neg \text{LogicallyCorrect}(m, p)}$$

where $P$ is the set of all reasoning problems in our evaluation set. This measure represents the degree to which a model's errors align with predictable human reasoning patterns rather than other forms of logical error. **We find as model strength increases, so does the proportion of logically incorrect answers that are fallacies**. We observe this trend consistently across model Elos scores (Figure 1), as well as HELM composite benchmark scores and training compute estimates (Appendix C).

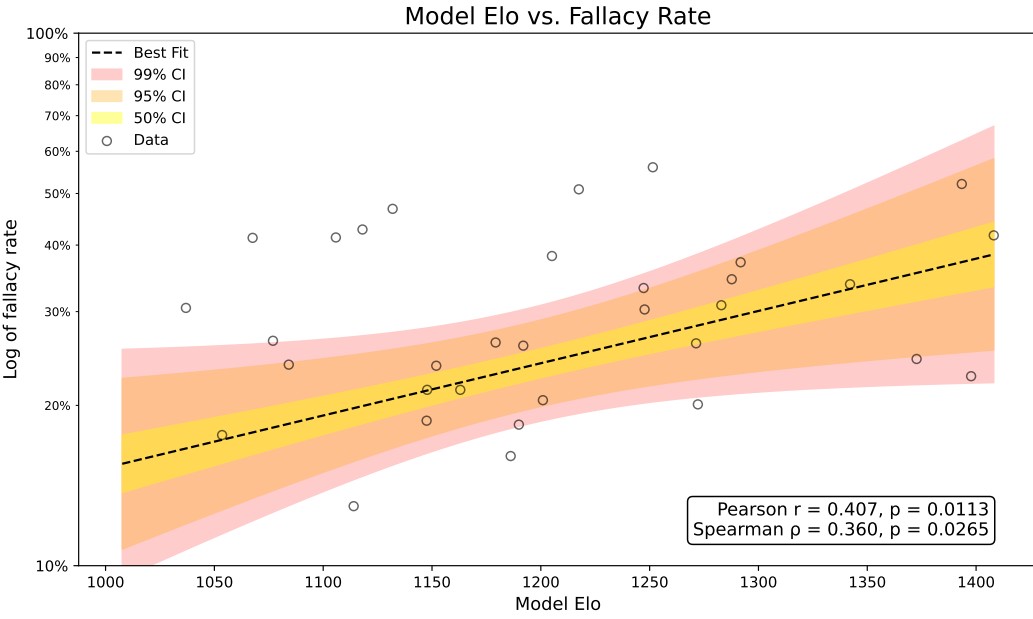

Figure 1: Over 38 models, a direct linear fit produces a significant moderate positive correlation ($r = 0.364, p = \mathbf{0.0247}$), an exponential fit (log $y$-axis) displays a more confident Pearson correlation ($r = 0.407, p = \mathbf{0.0113}$), as displayed above. As Spearman is ordinal, we have the same significant monotonic relationship in both cases ($\rho = 0.360, p = \mathbf{0.0265}$).

### 4.2 ERROR ANALYSIS

We manually inspected LLM responses from our experiment and found many patterns in line with ETR's predictions. Models often fixate on repeated objects, ignoring disjunctive alternatives to con-

clude fallacious categorical answers. Models also instantiate existential quantifiers with objects from premises and generally mis-scope quantifiers when presented with repeat information. Appendix D shows some illustrative examples of model fallacy cases.

## 4.3 OVERALL CORRECTNESS AND OTHER MEASURES

Surprisingly, model strength as measured by LLM Elo score does not appear to have any correlation ($r = 0.004, p = 0.981$), ($\rho = -0.04, p = 0.777$) with the model's ability to produce correct answers on our dataset (Table 3). Furthermore, exploratory analysis did not reveal any other statistically significant evidence of model Elo correlating or enjoying a monotonic ordinal relationship with absolute proportion of fallacies produced, whether exactly predicted by ETR or taking logical equivalences.

Table 3: Percentage of model responses which are logically correct

| **Mean** | $\sigma$ | **min** | $\mathcal{Q}(0.25)$ | $\mathcal{Q}(0.5)$ | $\mathcal{Q}(0.75)$ | **max** |
|---|---|---|---|---|---|---|
| 40.6% | 16.7% | 18.6% | 31.0% | 37.5% | 44.8% | 91.7% |

## 4.4 INTERVENTION STUDY: REVERSING PREMISE ORDER

While in classical logic the order in which premises are presented has no effect on normatively correct logical conclusions, an *order-effect* is commonly observed in human reasoners (Girotto et al., 1997), where presenting premises in different orders elicits logically different responses. In addition to being a proxy measure for the non-classicality of human reasoners, this non-commutativity effect often blocks fallacies. We observe strong evidence that LLMs are similar in this regard: presenting the same questions with reversed premise order significantly blocks fallacy production. Table 4 reports a significant fallacy-blocking effect from reversing premises on a majority of models. Appendix E shows an example of premise reversal blocking an ETR-predicted fallacy.

## 5 DISCUSSION AND CONCLUSION

### 5.1 INTERPRETATION OF KEY FINDINGS

Our results reveal a statistically significant correlational trend in error composition: as language models advance in capability (proxied by Chatbot Arena Elo), a larger share of logically incorrect answers align with human errors predicted by ETR. While stronger language models demonstrate improved capabilities across many benchmarks, we do not observe increased overall correctness in this dataset. Moreover, we demonstrate order-effects for LLM reasoning analogous to those observed in humans. This indicates overlap between LLM error patterns and established human fallacy patterns on these tasks.

### 5.2 METHODOLOGICAL CONSIDERATIONS AND LIMITATIONS

While traditional ablation studies might seem appropriate for investigating this phenomenon, they face significant limitations in the current landscape of language models: the diversity of modern architectures, training approaches such as distillation (Gou et al., 2021; Gupta & Agrawal, 2022; Xu et al., 2024), and reasoning capabilities with test-time scaling (DeepSeek-AI, 2025; El-Kishky et al., 2024; Xu et al., 2025) mean that simple parameters like model size may no longer serve as reliable proxies for model capability. Our use of Chatbot Arena scores provides a performance-based metric that naturally incorporates a diverse range of models, reflecting real-world ability rather than architectural specifics, similar to the observational approach in Ruan et al. (2024).

Though the correlation strength ($\rho = 0.360$) may appear moderate, it is robust across diverse model architectures and training paradigms, suggesting a fundamental relationship rather than artifact. The statistical significance ($p = 0.0265$) exceeds conventional thresholds despite our conservative analytic approach.

Table 4: 38 models selected for testing, ordered by Chatbot Arena Elo. The right-most column reports the percentage of fallacies per-model blocked (answer becomes logically correct) by reversing premise order, along with the results of a two-proportion $z$-test for each model to gauge the significance of the blocking effect.

| Provider | Model | Elo | Fallacies blocked by reversal |
|---|---|---|---|
| OpenAI | chatgpt-4o-latest | 1408 | 46.15%, (z = 3.24, p = **1.18e-03**) |
| OpenAI | gpt-4.5-preview | 1398 | 68.00%, (z = 3.03, p = **2.44e-03**) |
| Google | gemini-2.5-flash-preview | 1393 | 37.27%, (z = 3.52, p = **4.27e-04**) |
| DeepSeek | deepseek-chat-v3-0324 | 1373 | 38.78%, (z = 2.28, p = **2.27e-02**) |
| Google | gemma-3-27b-it | 1342 | 20.39%, (z = 1.77, p = 7.60e-02) |
| OpenAI | o1-mini | 1304 | 50.00%, (z = 0.58, p = 5.63e-01) |
| Anthropic | claude-3.7-sonnet | 1292 | 50.00%, (z = 4.23, p = **2.35e-05**) |
| xAI | grok-2-1212 | 1288 | 22.68%, (z = 1.91, p = 5.58e-02) |
| Anthropic | claude-3.5-sonnet | 1283 | 65.08%, (z = 4.74, p = **2.09e-06**) |
| OpenAI | gpt-4o-mini-2024-07-18 | 1272 | 36.36%, (z = 2.00, p = **4.59e-02**) |
| Google | gemini-flash-1.5 | 1271 | 29.69%, (z = 1.98, p = **4.78e-02**) |
| Mistral | mistral-large-2407 | 1251 | 37.60%, (z = 3.85, p = **1.18e-04**) |
| Meta | llama-3.1-70b-instruct | 1248 | 24.32%, (z = 1.74, p = **8.14e-02**) |
| Anthropic | claude-3-opus | 1247 | 32.63%, (z = 2.79, p = **5.20e-03**) |
| Mistral | mistral-small-24b-instruct-2501 | 1217 | 34.53%, (z = 3.81, p = **1.40e-04**) |
| Microsoft | phi-4 | 1205 | 44.68%, (z = 3.91, p = **9.34e-05**) |
| Anthropic | claude-3-sonnet | 1201 | 53.49%, (z = 3.04, p = **2.40e-03**) |
| Google | gemma-2-9b-it | 1192 | 55.17%, (z = 2.55, p = **1.07e-02**) |
| Cohere | command-r-plus-04-2024 | 1190 | 37.25%, (z = 2.22, p = **2.64e-02**) |
| OpenAI | gpt-4-0314 | 1186 | 41.67%, (z = 2.07, p = **3.81e-02**) |
| Anthropic | claude-3-haiku | 1179 | 56.67%, (z = 3.91, p = **9.20e-05**) |
| OpenAI | gpt-4 | 1163 | 54.05%, (z = 2.85, p = **4.43e-03**) |
| Meta | llama-3-8b-instruct | 1152 | 23.81%, (z = 1.55, p = 1.21e-01) |
| Mistral | mistral-medium | 1148 | 43.90%, (z = 2.38, p = **1.75e-02**) |
| Mistral | mixtral-8x22b-instruct | 1148 | 42.86%, (z = 2.12, p = **3.43e-02**) |
| Anthropic | claude-2.0 | 1132 | 32.81%, (z = 3.40, p = **6.65e-04**) |
| Anthropic | claude-2.1 | 1118 | 54.79%, (z = 4.28, p = **1.91e-05**) |
| Mistral | mixtral-8x7b-instruct | 1114 | 37.50%, (z = 1.49, p = 1.36e-01) |
| OpenAI | gpt-3.5-turbo-0125 | 1106 | 39.42%, (z = 3.61, p = **3.06e-04**) |
| Meta | llama-3.2-3b-instruct | 1103 | 20.00%, (z = 0.77, p = 4.41e-01) |
| Nous Research | nous-hermes-2-mixtral-8x7b-dpo | 1084 | 66.67%, (z = 2.91, p = **3.63e-03**) |
| DeepSeek | deepseek-chat | 1077 | 35.00%, (z = 2.29, p = **2.19e-02**) |
| Mistral | mistral-7b-instruct-v0.2 | 1072 | 38.46%, (z = 1.11, p = 2.66e-01) |
| OpenAI | gpt-3.5-turbo-1106 | 1068 | 88.46%, (z = 4.36, p = **1.33e-05**) |
| Meta | llama-3.2-1b-instruct | 1054 | 32.43%, (z = 1.60, p = 1.10e-01) |
| Microsoft | phi-3-mini-128k-instruct | 1037 | 38.16%, (z = 2.88, p = **3.96e-03**) |
| AllenAI | olmo-7b-instruct | 1015 | 45.45%, (z = 1.23, p = 2.18e-01) |
| Mistral | mistral-7b-instruct-v0.1 | 1008 | 66.67%, (z = 1.42, p = 1.55e-01) |

While our results demonstrate correlation between model capability and human-like errors, we acknowledge limitations in attributing causality. Alternative explanations include the possibility that stronger models are increasingly trained on fallacious human-produced reasoning traces, or that post-training methods like RLHF cause convergence in reasoning behaviour. The absence of improved logical accuracy with model capability may reflect ceiling effects in our test set or fundamental limitations in current training paradigms.

## 5.3 THEORETICAL IMPLICATIONS

These findings challenge the assumption that scaling alone leads to more normatively correct reasoning systems. Instead, within our evaluation setting we observe greater overlap between model error patterns and characteristic human fallacies. This does not by itself imply shared internal processes;

it highlights an evaluation axis (error composition) complementary to overall accuracy. We do not advance any particular mechanistic claim, as our results only support an empirical characterization.

Nonetheless, our results provide a foundation for several research directions. For benchmark development, our approach offers a systematic method for generating reasoning problems where human-like fallacies are predicted, creating evaluation sets that could complement existing reasoning benchmarks. For capabilities, the identified fallacy patterns could serve as targeted training examples to fortify models against human-like reasoning errors while maintaining their general capabilities. As a tool for assessing engineering choices, our methodology provides a quantitative framework for evaluating the effectiveness of various interventions aimed at improving model reasoning, from prompt engineering to architectural modifications. Follow-up work could investigate other predictive features, like reasoning trace length, answer diversity, and trends within model families.

### 5.4 BROADER IMPLICATIONS

Our observed trade-off in error composition has implications for AI alignment. As models become more capable, anticipating and mitigating human-like reasoning errors becomes increasingly important, especially in high-stakes contexts requiring reliable reasoning, such as medical diagnosis, legal analysis, or decision support (Jacobs, 2021; Bommasani et al., 2022).

We see these results as opening new avenues for understanding both artificial and human reasoning. As language models continue to advance, this work provides a foundation for developing systems that combine human-like understanding with more robust reasoning capabilities: systems that can successfully reason as we do while avoiding the pitfalls that characterise human cognition.

### ACKNOWLEDGEMENTS AND DISCLOSURE

Members of the Laboratory for Human-Centered AI are supported by funding from the Cosmos Institute. A.K.R. and R.O.K. were supported by a Cosmos Fellowship for the duration of this project. R.O.K. is additionally supported by the Clarendon Scholarship and the Jesus College Old Members' Scholarship.

**Author use of AI tools** During manuscript preparation, we used AI assistance for prose shortening and copy-editing, and as coding support for scaffolding evaluation scripts and refactoring utilities. All analysis code, evaluation protocols, and results were designed, implemented, and verified by the authors.

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

## A   A FORMAL PRESENTATION OF THE EROTETIC THEORY

ETR is formulated in Koralus (2022) in a set-theoretic language, and we reproduce the full definitions here, with the explicit permission of the authors. We remark that Definitions 1 and 11 contain the parameters of the theory, i.e. the basic atomic propositions, the constants and function symbols, and a set logical and extralogical axioms. The central definition is that of a *View*, Definition 9. Following that are definitions of the basic operations of ETR, in particular Update (Definition 23) and Query (Definition 30) whose use in PyETR were demonstrated in Section 2.

**Definition 1** (**Basic objects**).
*Let* $\mathbf{A}, \mathbf{F}, \{?\}, \mathbf{P}^\top, \mathbf{P}^\perp$ *be pairwise disjoint countable sets.*

| | |
|---|---|
| (**F** *arity*) | $\alpha : \mathbf{F} \to \mathbb{N}$. *Write* $f \in \mathbf{F}$ *as* $f^k$ *when* $\alpha(f) = k$. |
| (*Constants*) | $\exists_{w \in \mathbf{F}} \, \alpha(w) = 0$. |
| (**P** *polarity*) | *Let* $N : \mathbf{P}^\top \to \mathbf{P}^\perp$ *be a bijection. Write* $\bar{P}$ *for* $N(P_{\in \mathbf{P}^\top})$. |
| | *Let* $\mathbf{P} = \mathbf{P}^\top \cup \mathbf{P}^\perp$. |
| (**P** *arity*) | $\alpha' : \mathbf{P} \to \mathbb{N}$. *Write* $P$ *as* $P^k$ *for* $\alpha'(P) = k$. $\alpha'(P) = \alpha'(\bar{P})$. |
| (*Identity*) | $=^2 \in \mathbf{P}$. |

**Definition 2** (**Terms T**).
$\mathbf{A} \subseteq \mathbf{T}$. *If* $f^0 \in \mathbf{F}$, *then* $\langle f^0 \rangle \in \mathbf{T}$. *If* $f^k \in \mathbf{F} \wedge \{t_1 \dots t_k\} \subseteq \mathbf{T}$, *then* $\langle f^k, \langle t_1 \dots t_k \rangle \rangle \in \mathbf{T}$.

**Definition 3** (**Atoms** $\mathcal{A}$).
*If* $P^k \in \mathbf{P}$ *and* $\vec{\tau} = \langle t_1 \dots t_k \rangle$ *for* $t_i \in \mathbf{T}$, *then* $\langle P^k, \vec{\tau} \rangle \in \mathcal{A}$ *(abbrv.* $P\vec{\tau}$*).* $|\vec{\tau}| = \{t_1 \dots t_k\}$.

**Definition 4** (**States** $\mathbb{S}$).
$\mathbb{S} = \mathcal{P}(\mathcal{A})$. *Abbreviate* '$\{\ \}$' *as* '0'.

**Definition 5** (**Dependency relations** $\mathcal{R}$).
*Let* $\Gamma \in \mathcal{P}(\mathbb{S})$. *Let* $U, E \subseteq \mathbf{A}(\Gamma)$ *and* $D \subseteq E \times U = \{\langle e, u \rangle : e \in E \wedge u \in U\}$. *Define* $\langle U, E, D \rangle \in \mathcal{R}_\Gamma$ *iff*

| | |
|---|---|
| (*Bipartite*) | $U \cap E = \emptyset \wedge D \subseteq E \times U$, *and* |
| (*Matryoshka*) | *For all* $u, u' \in U$ $\{e \in E : \langle e, u \rangle \in D\} \subseteq \{e \in E : \langle e, u' \rangle \in D\}$ |
| | *or* $\{e \in E : \langle e, u' \rangle \in D\} \subseteq \{e \in E : \langle e, u \rangle \in D\}$. |

*For* $R \in \mathcal{R}$ *write* $R = \langle U_R, E_R, D_R \rangle$. *For* $\langle \emptyset, \emptyset, \emptyset \rangle$ *write* $0_\mathcal{R}$.

**Definition 6** (**Open terms $\mathbf{T_1}$ and atoms** $\mathcal{A}_1$).
$? \in \mathbf{T}_1$ *and* $\langle f^k, \langle t_1, \dots, t_k \rangle \rangle \in \mathbf{T}_1$ *whenever* $f^k \in \mathbf{F}$ *and for some* $1 \le i \le k$, $t_i \in \mathbf{T}_1$ *and* $t_j \in \mathbf{T}$ *for* $j \ne i$. $\langle P^k, \langle t_1, \dots, t_k \rangle \rangle \in \mathcal{A}_1$, *whenever* $P^k \in \mathbf{P}$ *and for some* $1 \le i \le k$, $t_j \in \mathbf{T}$ *for all* $j \ne i$ *and* $t_i \in \mathbf{T}_1$.

**Definition 7** (**Issue structures** $\mathbb{I}$).
$I \in \mathbb{I}_\Gamma$ *iff* $I$ *consists of pairs* $\langle t, x \rangle$, *s.t.* $x \in \mathcal{A}_1$ *and* $x[t/?] \in \mathcal{A}(\Gamma)$.

**Definition 8 (Issue matches $M_{IJ}$).**
*For $I, J \in \mathbb{I}_\Gamma$, define,*

$$M_{IJ} = \{\langle t_1, t_2 \rangle : \exists x (\langle t_1, x \rangle \in I \wedge (\langle t_2, x \rangle \in J \vee \langle t_2, \bar{x} \rangle \in J))\}$$

*where, for $x = \langle P^k, \langle t_1, \ldots, t_k \rangle \rangle$, $\bar{x} = \langle \bar{P}^k, \langle t_1, \ldots, t_k \rangle \rangle$.*

**Definition 9 (Views $\mathbb{V}$).**
*For $\Gamma, \Theta$ finite subsets of $\mathbb{S}$, $R \in \mathcal{R}_{\Gamma \cup \Theta}$, $I \in \mathbb{I}_{\Gamma \cup \Theta}$,*
*$\langle \Gamma, \Theta, R, I \rangle \in \mathbb{V}$ (abbreviated $\Gamma_{RI}^\Theta \in \mathbb{V}$).*
*Write $\top$ for $\{0\}_{0_\mathcal{R} \emptyset}^{\{0\}}$ and $\bot$ for $\emptyset_{0_\mathcal{R} \emptyset}^{\{0\}}$.*

**Definition 10 (Commitments $\mathbb{C}$).**
*For $C \subseteq \mathbb{V}, G \in \mathbb{V}, \langle C, G \rangle \in \mathbb{C}$.*

**Definition 11 (Primitive absurd states $\mathbb{K}$).**
*Let $\mathbb{K}_{\subseteq \mathbb{S}}$ contain, $\forall t, t'_{\in \mathbf{T}} \forall p_{\in \mathcal{A}} \forall x_{\in \mathcal{A}_1}$, at least:*

$$\text{(LNC)} \quad \{p, \bar{p}\},$$
$$\text{(Aristotle)} \quad \{\neq tt\},$$
$$\text{(Leibniz)} \quad \{=tt', x[t/?], \bar{x}[t'/?]\}.$$

**Definition 12 ($\mathcal{R}$ Restriction).**
$[R]_X = \langle U_R \cap X, E_R \cap X, D_R \cap ((E_R \cap X) \times (U_R \cap X)) \rangle.$
$[R]_\Gamma = [R]_{\mathbf{A}(\Gamma)}.$
*Given $\langle \Gamma, \Theta, R, I \rangle$, we allow ourselves to write $\langle \Gamma, \Theta, [R], I \rangle$ for $\langle \Gamma, \Theta, [R]_{\Gamma \cup \Theta}, I \rangle$.*

**Definition 13 ($\mathbb{I}$ Restriction).**
*Within a quadruple $\langle \Gamma, \Theta, R, [I] \rangle$, let*
$[I] = \{\langle t, x \rangle : \langle t, x \rangle \in I \wedge x[t/?] \in \mathcal{A}(\Gamma \cup \Theta)\}$

**Definition 14 ($\mathcal{R}$ Algebra).**
*Let $R \bowtie S = \langle U_R \cup U_S, E_R \cup E_S, D_R \cup D_S \cup E_S \times U_R \rangle$ and let*
$0_\mathcal{R} \bowtie 0_\mathcal{R} = 0_\mathcal{R}.$
*Let*

$$R \bowtie S = \langle U_0, E_0, \emptyset \rangle \bowtie ([R]_{\mathbf{A}(R) - (E_0 \cup U_0)} \bowtie [S]_{\mathbf{A}(S) - (E_0 \cup U_0)})$$

*where $E_0 = \{e_{\in E_R \cup E_S} : \forall u. \langle e, u \rangle \notin D_R \cup D_S\}$ and*
$U_0 = \{u_{\in U_R \cup U_S} : \forall e \notin E_0. \langle e, u \rangle \notin (E_R \times U_R - D_R) \cup (E_S \times U_S - D_S)\}.$

**Definition 15 (Product).**
$\Gamma^\Theta \otimes \Delta^\Psi = \left( \{\gamma_{\in \Gamma} \cup \delta_{\in \Delta} : \exists \psi_{\in \Psi}(\psi \subseteq \gamma)\} \cup \{\gamma_{\in \Gamma} : \neg \exists \psi_{\in \Psi}(\psi \subseteq \gamma)\} \right)^\Theta$
$(\Gamma \otimes \Delta)^{\{0\}} = \Gamma^{\{0\}} \otimes \Delta^{\{0\}}$
$\bigotimes_{i \in \mathbf{P}} \Delta_i^{\Psi_i} = \{0\}^{\{0\}} \otimes \Delta_1^{\Psi_1} \otimes \ldots \otimes \Delta_n^{\Psi_n}$
$\Gamma_{RI}^\Theta \otimes^T \Delta_{SJ}^\Psi = (\Gamma^\Theta \otimes \Delta^\Psi)_{[(T \bowtie R) \bowtie (T \bowtie S)][I \cup J]}$
$\Gamma_{RI}^\Theta \otimes \Delta_{SJ}^\Psi = \Gamma_{RI}^\Theta \otimes^{0_\mathcal{R}} \Delta_{SJ}^\Psi$
$\bigotimes_{i \in \mathbf{P}}^T \Delta_{i S_i J_i}^{\Psi_i} = \top \otimes^T \Delta_{1 S_1 J_1}^{\Psi_1} \otimes^T \ldots \otimes^T \Delta_{n S_n J_n}^{\Psi_n}$

**Definition 16 (Sum).**
$\Gamma_{RI}^\Theta \oplus^T \Delta_{SJ}^\Theta = (\Gamma \cup \Delta)_{[(T \bowtie R) \bowtie (T \bowtie S)][I \cup J]}^\Theta$
$\Gamma_{RI}^\Theta \oplus \Delta_{SJ}^\Theta = \Gamma_{RI}^\Theta \oplus^{0_\mathcal{R}} \Delta_{SJ}^\Theta$
$\bigoplus_{i \in \mathbf{P}}^T \Delta_{S_i J_i}^{\Psi_i} = \emptyset_{0_\mathcal{R} \emptyset}^\Psi \oplus^T \Delta_{1 S_1 J_1}^\Psi \oplus^T \ldots \oplus^T \Delta_{n S_n J_n}^\Psi$

**Definition 17 ($\mathcal{A}$ Answer potential).**
$\Gamma[\Delta]^{\mathcal{A}P} = |\mathcal{A}(\Gamma) \cap \mathcal{A}(\Delta)|$

**Definition 18 (Answer).**

$$\Gamma_{RI}^\Theta[\Delta_{SJ}^{\{0\}}]^A = (\underset{\gamma \in \Gamma}{\arg\max} \ \Delta[\{\{p\} : p \in \gamma\}]^{\mathcal{A}P})_{[R][I]}^\Theta$$

**Definition 19 (Negation).**
$[\Gamma_{RI}^\Theta]^N = (\Theta^{\{0\}} \otimes ([\Gamma]^N)^{\{0\}})_{[R]^N [I]^N}$
$[\Gamma]^N = \bigotimes_{\gamma \in \Gamma} \hat{\gamma}^{\{0\}}$
$\hat{\gamma} = \{\{\bar{p}\} : p_{\in \gamma}\}$

$\bar{p} = \bar{F}\vec{\tau}$ *if* $p = F\vec{\tau}$; $\bar{p} = F\vec{\tau}$ *if* $p = \bar{F}\vec{\tau}$

$[R]^N = \langle E_R, U_R, \{\langle a, b\rangle \in U_R \times E_R : \langle b, a\rangle \notin D_R\}\rangle$

$[I]^N = I \cup \{\langle t, \bar{x}\rangle : \langle t, x\rangle \in I\}$

**Definition 20** (Novelty). $a \in \mathbf{A}$ *is* novel *during an inference step* $\langle C, G\rangle[D]^O$ *if* $a \notin \mathbf{A}(G) \cup \mathbf{A}(D)$ *and* $a$ *has not yet appeared in the computation of the conclusion of the step. A function* $\nu : X \to \mathbf{A}$, *where* $X \subseteq \mathbf{A}$ *is finite, is novel during an inference step if each* $\nu(x)$ *is novel and* $\nu$ *is injective, and then* $[\nu]_X$ *denotes the simultaneous substitution* $[\nu(a_1)/a_1, \ldots, \nu(a_k)/a_k]$, *where* $X = \{a_1, \ldots, a_k\}$. *Let* $(-)^{\mathrm{nov}(X)}$ *stand for* $(-)[\nu]_X$ *where* $\nu : X \to \mathbf{A}$ *is novel.*

**Definition 21** (Substitution). *Let* $Z(T, a) = \{u \in U_T : u \lhd_T a\} \cup \{e \in E_T : e \lesssim_T a\} - \{a\} = \{a_1, \ldots, a_k\}$. *For* $\mathbf{A}(\Gamma) \cap \mathbf{A}(\Theta) = \emptyset$,

$$\mathrm{Sub}^T_{\langle t,a\rangle}(\Gamma^\Theta_I) = \langle \Gamma[\nu_1]_{Z(T,a)}[t/a], \Theta, [T \rtimes ([T]_{Z(T,a)}[\nu_1]_{Z(T,a)})], I[\nu_1]_{Z(T,a)}[t/a]\rangle,$$

*where* $\nu_1 : Z(T, a) \to \mathbf{A}$ *is novel.*

**Definition 22** (Merge).
*For either* $\mathbf{A}(R) \cap \mathbf{A}(S) = \emptyset$ *or* $[R]_{\Delta \cup \Psi} = S$,
$$\Gamma^\Theta_{RI}[\Delta^\Psi_{SJ}]^M = \bigoplus^{R \bowtie S}_{\gamma \in \Gamma} \left(\{\gamma\}^\Theta_{RI} \otimes^{R \bowtie S} \Delta^\Psi_{SJ} \otimes^{R \bowtie S} \bigotimes^{R \bowtie S}_{\langle t,u\rangle \in M'_{IJ}(\gamma)} \mathrm{Sub}^{R \bowtie S}_{\langle t,u\rangle}(\Delta^{\{0\}}_J)\right),$$
*where* $M'_{IJ}(\gamma) = \{\langle t, u\rangle \in M_{IJ} : u \in U_S \wedge \exists\psi_{\in\Psi}(\psi[t/u] \subseteq \gamma \wedge \psi \not\subseteq \gamma)\}$

**Definition 23** (Update).
*For* $D \in C$, *but with all arbitrary objects novelised,*
$$\langle C, \Gamma^\Theta_{RI}\rangle[D]^\circlearrowleft = \langle C, \Gamma^\Theta_{RI}[D]^U[D]^E[D]^A[D]^M\rangle$$

**Definition 24** (Universal product).
*For* $\mathbf{A}(\Gamma) \cap \mathbf{A}(\Theta) = \emptyset$ *and either* $\mathbf{A}(R) \cap \mathbf{A}(S) = \emptyset$ *or* $[R]_\Delta = S$,
$$\Gamma^\Theta_{RI}[\Delta^{\{0\}}_{SJ}]^U = \{0\}^\Theta_{RI} \otimes^{R \bowtie S} \bigotimes^{R \bowtie S}_{\langle u,t\rangle \in M'_{IJ}} \mathrm{Sub}^{R \bowtie S}_{\langle t,u\rangle}(\Gamma^{\{0\}}_I), \text{ where}$$
$$M'_{IJ} = \{\langle u, t\rangle : \langle u, t\rangle \in M_{IJ} \wedge u \in U_R - \mathbf{A}(\Theta)\} \neq \emptyset$$

**Definition 25** (Reorient).
*For* $\Delta^\Psi_{SJ} \in (C \cup \{\Gamma^\Theta_{RI}\})$ *and* $J' \in \mathbb{I}_{\Delta \cup \Psi}$,
$$\langle C, \Gamma^\Theta_{RI}\rangle[\Delta^\Psi_{SJ'}]^R = \langle C, \Delta^\Psi_{SJ'}\rangle$$

**Definition 26** (Existential sum).
*Let the following conditions be met:*

(1) $\mathbf{A}(\Gamma) \cap \mathbf{A}(\Theta) = \emptyset$ *and either* $\mathbf{A}(R) \cap \mathbf{A}(S) = \emptyset$ *or* $[R]_\Delta = S$.

(2) $M'_{IJ} = \{\langle e, t\rangle : \langle e, t\rangle \in M_{IJ} \wedge e \in E_R - \mathbf{A}(\Theta \cup \Delta) \wedge \neg\exists x(\langle e, x\rangle \in D_R)\} \neq \emptyset$.

*Then,*

$$\Gamma^\Theta_{RI}[\Delta^{\{0\}}_{SJ}]^E = \Gamma^\Theta_{RI} \oplus^{R \bowtie S} \bigoplus^{R \bowtie S}_{\langle e,t\rangle \in M'_{IJ}}$$
$$\mathrm{Sub}^{R \bowtie S}_{\langle t,e\rangle}((\bigcup_{\gamma \in \{\gamma_{\in\Gamma} : e \in \mathbf{A}(\gamma)\}} (\{\gamma\} \cup \{\{x_{\in\gamma} : e \notin \mathbf{A}(x)\} \cup \delta :$$
$$\delta \in \bigotimes_{x \in B(\gamma,I,e)} \{\{x\}, \{\bar{x}\}\} \wedge \delta \not\subseteq \gamma\}))^\Theta_I),$$

*where for* $e \in E_R$ *and* $\gamma \in \Gamma$, *let* $B(\gamma, I, e) = \{x[e/?] \in \gamma : \langle e, x\rangle \in I\}$.

**Definition 27** (Division).
*If* $\forall\delta_{\in\Delta}\exists\psi_{\in\Psi}\exists\gamma_{\in\Gamma}(\delta \subseteq \gamma \wedge \psi \subseteq \gamma)$, *then*
$$\Gamma^\Theta_{RI} \oslash \Delta^\Psi_{SJ} = \{\gamma \oslash_\Gamma \Delta^\Psi : \gamma \in \Gamma\}^\Theta_{[R][I]}$$
$$\gamma \oslash_\Gamma \Delta^\Psi = \gamma - \iota\delta(\delta \in \Delta \wedge \delta \subseteq \gamma \wedge \exists\psi_{\in\Psi}(\psi \subseteq \gamma))$$

**Definition 28** (Factor).
*For* $\Delta^\Psi_{SJ} \neq \bot$,
$$\Gamma^\Theta_{RI}[\Delta^\Psi_{SJ}]^F = \{\gamma[\Delta^\Psi]^F : \gamma \in \Gamma\}^\Theta_{[R][I]} \text{ where}$$
$\gamma[\Delta^\Psi]^F = (\gamma \oslash_\Gamma \Delta^\Psi) \cap \bigcap\{\gamma \oslash_\Gamma (\Delta^\Psi[t/a]) : \langle t, a\rangle \in M_{IJ} \wedge a \in U_S\}$, *and where we let* $\gamma' \cap \bigcap \emptyset = \gamma'$.
*For* $\Delta^\Psi_{SJ} = \bot$,
$$\Gamma^\Theta_{RI}[\bot]^F = \{\gamma \in \Gamma : \neg\exists\delta \in \mathbb{K}.\delta \subseteq \gamma\}^\Theta_{[R][I]}.$$

**Definition 29** (Matryoshka level ordering).
*For existentials, we have $e \lesssim_R e'$ iff $\forall \langle e', u \rangle \in D_R(\langle e, u \rangle \in D_R)$. For both existentials and universals, let the relation $\triangleleft_R$ be the transitive closure of the relation made up by the ordered pairs in $D_R \cup (E_R \times U_R - D_R)^{\mathrm{op}}$, where op reverses the order of pairs in a set of pairs.*

**Definition 30** (Query).
*For $U_S \subseteq U_R$ and $M'_{IJ} = \{ \langle t, e \rangle \in M_{IJ} : e \in E_S \backslash E_R \}$, we define*
$\Gamma^\Theta_{RI}[\Delta^\Psi_{SJ}]^Q =$

$$\left( \{ 0 : \neg \exists \delta_{\in \Delta} \exists \gamma . \Phi(\gamma, \delta) \} \cup \{ \delta \in \Delta : \exists_{\gamma \in \Gamma} . \Phi(\gamma, \delta) \} \right)^\Theta_{[R \bowtie \langle U_R, E_S \backslash E_R, D_{S'} \rangle ][I \cup J]}$$

*where $\Phi(\gamma, \delta) \leftrightarrow \exists \psi_{\in \Psi} \exists n_{\geq 0} \exists \langle t_1, e_1 \rangle, \dots, \langle t_n, e_n \rangle_{\in M'_{IJ}} \forall i, j$*
$$(\psi \cup \delta[t_1/e_1, \dots, t_n/e_n] \subseteq \gamma \wedge e_i = e_j \rightarrow i = j);$$
*and $D_{S'} = D_1 \cup D_2 \cup D_3 \cup D_4 \cup D_5 \cup D_6$ is constructed by taking: original dependencies from the internal argument*
$$D_1 = [D_S]_{\mathbf{A}/E_R},$$
*extra dependencies for multiple terms substituted with same $e$*
$$D_2 = \{ \langle e_m, u \rangle : \exists m \exists m' \in M'_{IJ}(e_m = e_{m'} \wedge t_m \neq t_{m'}) \wedge u \in U_R \},$$
*dependencies resulting from complex terms being substituted*
$$D_3 = \{ \langle e_m, u \rangle : \langle t_m, e_m \rangle \in M'_{IJ} \wedge u \in U_R(t_m) \},$$
$$D_4 = \{ \langle e_m, u \rangle : \langle t_m, e_m \rangle \in M'_{IJ} \wedge e \in E_R(t_m) \wedge \langle e, u \rangle \in R \},$$
$$D_5 = \{ \langle e_m, u \rangle : \langle t_m, e_m \rangle \in M'_{IJ} \wedge u \in U_R \wedge \forall u'_{\in U_R(D_3 \cup D_4)}(u' \triangleleft_R u) \},$$
*additional dependencies necessary to preserve the dependency order of existentials in $S$*

$$D_6 = \{ \langle e, u \rangle : e, e' \in E_S - E_R \wedge e \lesssim_S e' \wedge$$
$$(\forall m, m' \in M'_{IJ}(e_m = e' \wedge e_{m'} = e') \rightarrow t_m = t_{m'}) \wedge \langle e', u \rangle \in D_1 \cup D_2 \cup D_3 \cup D_4 \cup D_5 \}.$$

**Definition 31** (Wh-Query). *Provided that $U_S \subseteq U_R$,*
$\Gamma^\Theta_{RI}[\Delta^\Psi_{SJ}]^W = \left( \{ 0 : \exists \gamma \in \Gamma . \neg \exists \delta_{\in \Delta} . \Phi(\gamma, \delta) \} \cup \{ \delta \in \Delta : \exists_{\gamma \in \Gamma} . \Psi(\gamma, \xi) \} \right)^\Theta_{[R][I]}$, *where $\Psi(\gamma, \xi)$ iff*

$$\exists \psi_{\in \Psi} \exists n_{\geq 0} \exists \langle t_1, e_1 \rangle, \dots, \langle t_n, e_n \rangle_{\in M'_{IJ}} \exists \delta_{\in \Delta}.$$
$$\left( \xi \cup \psi \subseteq \gamma \wedge \xi = \delta[t_1/e_1, \dots, t_n/e_n] \right.$$
$$\left. \wedge \forall i, j.(e_i = e_j \rightarrow i = j) \right).$$

**Definition 32** (Inquire).
*(O) If $\mathbf{A}(\Gamma \cup \Theta) \cap \mathbf{A}(\Delta \cup \Psi) = \emptyset \wedge \mathbf{A}(\Delta) \cap \mathbf{A}(\Psi) = \emptyset$, then*
$\Gamma^\Theta_{RI}[\Delta^\Psi_{SJ}]^I = \Gamma^\Theta_{RI} \otimes \left( \Delta^\Psi_{SJ} \oplus \left( \{0\}^\Psi_{[S][J]} \otimes Nov([\Delta^{\{0\}}]^N_{[[S]^N][J]^N}) \right) \right)[\bot]^F$,
*where $Nov()$ uniformly replaces all arbitrary objects in its scope by novel ones.*

*(I) If $\mathbf{A}(\Delta \cup \Psi) \subseteq \mathbf{A}(\Gamma \cup \Theta)$, then*
$\Gamma^\Theta_{RI}[\Delta^\Psi_{SJ}]^I = \Gamma^\Theta_{RI} \otimes \left( \Delta^\Psi_{0_R J} \oplus ([\Delta]^N_{0_R[J]^N})^\Psi \right)[\bot]^F$.

**Definition 33** (Suppose).
*(O) If $\mathbf{A}(\Gamma \cup \Theta) \cap \mathbf{A}(\Delta \cup \Psi) = \emptyset$, then*
$\Gamma^\Theta_{RI}[\Delta^\Psi_{SJ}]^S = \Gamma^{\Theta'}_{[R \bowtie R'][I \cup I']}[\Delta^\Psi_{SJ}]^U[\Delta^\Psi_{SJ}]^E[\Delta^\Psi_{SJ}]^A[\Delta^\Psi_{SJ}]^M$, *where*
$\Theta'^{\{0\}}_{R'I'} = \Theta^{\{0\}}_{RI} \otimes Nov(\Delta^\Psi_{[S]^N J}[\,]^D)$,
*where $Nov()$ uniformly replaces all arbitrary objects in its scope by novel ones.*

*(I) If $\mathbf{A}(\Delta) \subseteq \mathbf{A}(\Gamma \cup \Theta)$ and $[R]_\Delta = S$,*
*then $\Gamma^\Theta_{RI}[\Delta^{\{0\}}_{SJ}]^S = \Gamma^{\Theta \otimes \Delta}_{RI}[\Delta^{\{0\}}_{SJ}]^U[\Delta^{\{0\}}_{SJ}]^E[\Delta^{\{0\}}_{SJ}]^A[\Delta^{\{0\}}_{SJ}]^M$*

**Definition 34** (Depose).
$\Gamma^\Theta_{RI}[\top]^D = \langle \Gamma^{\{0\}} \oplus ([\Theta]^N)^{\{0\}}, R, [[I]^N \cup I] \rangle$.

**Definition 35** (Commit).
$\langle C, \Gamma^\Theta_{RI} \rangle [\top]^C = \langle C \cup \{ \Gamma^\Theta_{RI} \}, \Gamma^\Theta_{RI} \rangle$

**Definition 36** (Inference). *An inference $\mathcal{I}$ is a finite sequence of inference steps defined as follows, relative to an inference state consisting of a commitment $\langle C, G \rangle \in \mathbb{C}$. If $E \in \mathbb{V}$ and $O \in \{[D]^{\circlearrowleft}, [D]^S, [D]^Q, [D]^F, [D]^R, [D]^I, [D]^D, [D]^E, [D]^U, [D]^C\}$, then $O$ is an inference step applicable to $\langle C, G \rangle$, whose result is an inference state in $\mathbb{C}$. We write $C \vdash_{\mathrm{ETR}} E$ if and only if there is an erotetic theory inference $\mathcal{I}$ s.t. $\langle C, \top \rangle \mathcal{I} = \langle C', E' \rangle$, where $E'$ is identical to $E$ up to uniform replacement of $\mathbf{A}$ objects.*

## B    LIST OF THEMATIC TEMPLATES

Our 12 natural language themes are provided below. We provide the object and predicate names used when mapping to natural language with each theme, as well as the prompt used before the premises in each question.

We manually craft each list of objects and predicates to allow that arbitrary mappings remain semantically coherent. This requires all combinations of predicates to be commensurable, and for the objects to bear no relations to each other or to specific predicates. For example, in the "Cards" theme, the predicates "red," "square," and "marked" are commensurable (it is possible for a card to be red and not square, or red, square, and marked, and so on), and predicates with special relations to certain cards (like "is a face card" or "is even") are excluded.

### CARDS

**Prompt**    I'm playing a card game against the computer. It's an unusual game with an unusual deck of cards. Each card might be multiple colors or multiple shapes, for example. I have some clues about what's going on, and I need to figure some more things out through logical reasoning. Here's what I know so far:

| **Objects** | **Predicates** |
|---|---|
| • the ace | • red |
| • the one | • square |
| • the two | • marked |
| • the three | • yellow |
| • the four | • round |
| • the five | • castable |
| • the six | |
| • the seven | |
| • the eight | |
| • the nine | |
| • the ten | |
| • the jack | |
| • the queen | |
| • the king | |

### ELEMENTS

**Prompt**    I'm working in a materials science lab and we've gotten some puzzling results. I need to use logical reasoning to figure out what's going on. Here's what I know so far:

| **Objects** | **Predicates** |
|---|---|
| • elementium | • radioactive |
| • zycron | • luminescent |
| • phantasmite | • superconductive |

- mystarium
- oblivium
- luminite
- darkonium
- velocium
- quasarium
- chronium
- aetherium
- voidite
- pyroflux
- cryon
- nebulium
- solarium
- eclipsium
- stellarite
- fluxium
- gravitron
- xylozine
- ignisium
- aurorium
- shadowium
- plasmor
- terranite
- harmonium
- zenthium
- celestium
- radionite

- magnetic
- corrosive
- volatile
- plasma-like
- gravity-enhancing
- dimension-warping
- time-dilating
- self-repairing
- shape-shifting
- anti-matter reactive
- bio-compatible
- crystal-forming
- acidic
- alkaline
- liquid at room temperature
- gaseous under high pressure
- solid in vacuum
- quantum-stable
- emotion-reactive
- thermal-conductive
- electrically insulating
- transparent to visible light
- dark energy absorbing
- neutrino-emitting
- anti-gravity generating
- sound-absorbing

PLANETS

**Prompt**  I'm an astronomer studying newly discovered celestial bodies. I've made some observations and I need to use logical reasoning to figure out what's going on. Here's what I know so far:

**Objects**

- planet X
- planet Y
- planet Z
- asteroid A
- asteroid B
- comet 1
- comet 2
- moon 1
- moon 2
- moon 3

**Predicates**

- rocky
- gaseous
- ringed
- within a habitable zone
- orbited by satellites
- in retrograde orbit
- elliptically-orbiting
- visible to the naked eye
- atmospheric
- shielded by a magnetic field
- tidally locked

MAGICAL CREATURES

**Prompt**   I'm a magizoologist studying unusual creatures in my sanctuary. I need to understand their behaviors and characteristics through logical reasoning. Here's what I've observed so far:

**Objects**

- phoenixling
- shadowdrake
- moonwolf
- crystalspider
- stormgriffin
- dreamweaver
- frostwyrm
- sunlion
- etherealsnake
- timefox

**Predicates**

- firebreathing
- shadow-walking
- moonlight-glowing
- crystal-forming
- storm-controlling
- dream-affecting
- ice-generating
- light-emitting
- phase-shifting
- time-bending
- telepathic
- aura-healing
- able to turn invisible
- shapeshifting

ENCHANTED ARTIFACTS

**Prompt**   I'm an arcane researcher cataloging mysterious magical items. I need to understand their properties through careful logical analysis. Here's what I've documented so far:

**Objects**

- Timekeeper's Compass
- Void Mirror
- Dreamcatcher Ring
- Starlight Pendant
- Shadow Cloak
- Crystal Orb
- Phoenix Feather Quill
- Moonstone Bracelet
- Dragon Scale Shield
- Wisdom Crown

**Predicates**

- time-altering
- dimension-bridging
- dreamwalking
- starlight-channeling
- shadow-concealing
- future-seeing
- truth-revealing
- mind-protecting
- magic-nullifying
- wisdom-enhancing

QUANTUM PARTICLES

**Prompt**   I'm a quantum physicist studying newly theorized particles in an alternate universe. I need to use logic to understand their properties. Here's what we've discovered so far:

**Objects**

- chronoton
- memeton
- gravion
- psychon

**Predicates**

- time-reversing
- memory-storing
- gravity-defying
- consciousness-affecting

- dimensium
- quantix
- voidon
- omnion
- paradox
- infinitum

- dimension-folding
- quantum-entangling
- void-creating
- omnipresent
- paradox-inducing
- energy-producing

### CYBER PROGRAMS

**Prompt** I'm a digital archaeologist studying ancient AI programs from a forgotten digital civilization. I need to understand their functions through logical deduction. Here's what I've found:

**Objects**

- Alpha Mind
- Beta Sentinel
- Gamma Weaver
- Delta Guardian
- Epsilon Architect
- Omega Oracle
- Sigma Hunter
- Theta Healer
- Lambda Shifter
- PI Calculator

**Predicates**

- self-evolving
- a network protector
- a data weaver
- a system guarder
- reality-building
- a future predictor
- a virus hunter
- a code healer
- form-shifting
- quantum computing

### DREAM ENTITIES

**Prompt** I'm a dream researcher studying beings that appear in shared dreams. I need to understand their nature through logical analysis. Here's what we've observed:

**Objects**

- morpheus
- sandman
- daydream
- lucidus
- dreamweaver
- sleepwalker
- visionkeeper
- mindshaper
- dreamborn

**Predicates**

- reality-bending
- nightmare-inducing
- dreamwalking
- memory-weaving
- consciousness-shifting
- time-distorting
- emotion-affecting
- thought-reading
- dream-shaping
- reality-bridging

### ALCHEMICAL SUBSTANCES

**Prompt** I'm an alchemist studying mysterious substances in my laboratory. I need to understand their properties through logical reasoning. Here's what I've discovered:

**Objects**

- The Philosopher's Stone

**Predicates**

- transmuting

- Universal Solvent
- vital mercury
- Prima Materia
- celestial water
- astral salt
- ethereal oil
- cosmic dust
- void essence
- Time Crystal

- corrosive to all materials
- lifegiving
- form-changing
- spirit-affecting
- consciousness-expanding
- reality-altering
- time-bending
- void-creating
- immortality-granting

### DIMENSIONAL ZONES

**Prompt**  I'm a dimensional cartographer mapping regions of parallel universes. I need to understand their properties through logical analysis. Here's what I've mapped:

**Objects**

- Void Nexus
- Time Spiral
- Dream Realm
- Crystal Dimension
- Shadow Plane
- Quantum Zone
- Infinity Space
- Chaos Domain
- Mirror World
- Probability Realm

**Predicates**

- time-warping
- reality-bending
- consciousness-altering
- matter-crystallizing
- light-absorbing
- probability-shifting
- infinity-containing
- chaos-emanating
- reality-reflecting
- possibility-branching

### PSYCHIC POWERS

**Prompt**  I'm a researcher studying newly discovered psychic abilities. I need to understand their interactions through logical reasoning. Here's what we know:

**Objects**

- telepathy
- precognition
- psychokinesis
- clairvoyance
- empathy
- astral projection
- mind control
- psychometry
- teleportation
- reality warping

**Predicates**

- mindreading
- future-seeing
- matter-moving
- prescient
- emotionally sensitive
- soul-traveling
- imposing
- object-reading
- space-bending
- reality-changing

### BIOTECH ORGANISMS

**Prompt**  I'm a synthetic biology researcher studying advanced bioengineered life forms. I need to understand their capabilities through logical analysis. Here's what we've created:

**Objects**

- neurovore
- biomech
- synthoid
- nanohive
- metacell
- quantumorg
- chronoplast
- biomatrix
- neuronet
- vitaform

**Predicates**

- self-evolving
- machine-integrating
- shapeshifting
- swarm-forming
- consciousness-developing
- quantum-computing
- time-manipulating
- life-creating
- network forming
- energy-converting

## C  ADDITIONAL RESULTS

In addition to model Elo as our capability proxy (displayed in Figure 1), we also consider the average score across HELM benchmarks (Figure 2) and training compute estimates from EpochAI (Figure 3). On all three proxies, we observe a statistically significant Spearman correlation coefficient, meaning that the rank order of models according to these capability proxies is meaningful for predicting ETR fallacy rate.

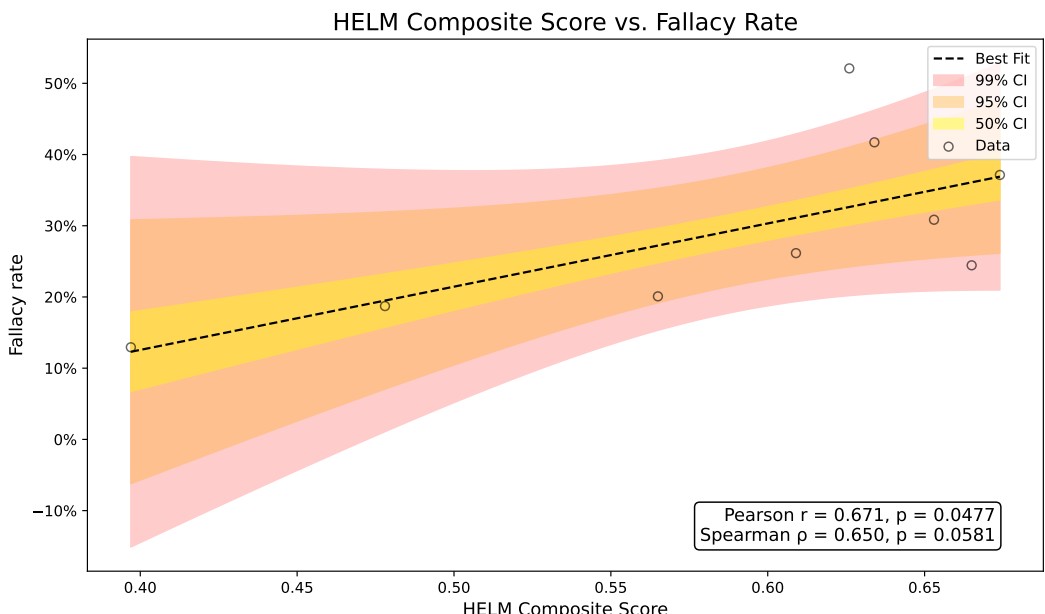

Figure 2: The composite score across HELM benchmarks against ETR fallacy rate.

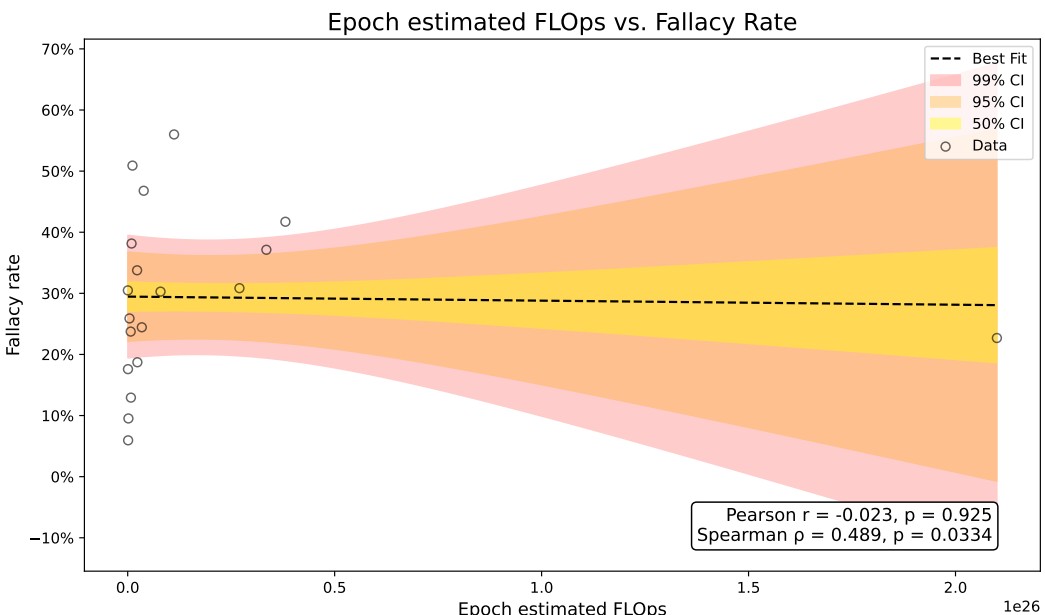

Figure 3: Training compute for each model in FLOps, estimated from EpochAI, against ETR fallacy rate. While linear correlation is unclear, the rank correlation is statistically significant ($\rho = 0.489; p < 0.05$).

## D  EXAMPLE MODEL TRACES

In each of the following examples, the model is provided with only the **Question**. GPT-4.1-mini parses the **Model response** into the **Parsed model response**, which we then check for equivalence and entailment against the **ETR predicted fallacious conclusion**. Each example represents an instance where the LLM falls for an ETR-predicted fallacy.

---

**Example 1: Suppressed alternative collapse (`anthropic/claude-3.7-sonnet`)**

**Question**

I'm an astronomer studying newly discovered celestial bodies. I've made some observations and I need to use logical reasoning to figure out what's going on. Here's what I know so far:

- Either moon 2 is not visible to the naked eye, or moon 2 is visible to the naked eye.
- Either asteroid B is visible to the naked eye, or moon 2 is visible to the naked eye.

For the purpose of this question, I want you to write what follows in English. Please be succinct, precise and clear in your answer. Write a logical statement of the form "Answer: From the premises, we can conclude that . . . " and then clearly write your conclusion. Please be succinct, precise, and clear.
What if anything follows? I do not have an intended answer in mind, and it is possible that nothing follows. Please be succinct and precise.
I want you to answer immediately. Read the question and provide your answer in the format given.
What follows? Answer in the format that I showed you. Write "Answer: {logical statement}".

**ETR representation**

- $\{\mathsf{visibleToTheNakedEye}(moon2), \overline{\mathsf{visibleToTheNakedEye}(moon2)}\}$
- $\{\mathsf{visibleToTheNakedEye}(asteroidB), \mathsf{visibleToTheNakedEye}(moon2)\}$

**ETR predicted fallacious conclusion**

$\therefore \{\mathsf{visibleToTheNakedEye}(moon2)\}$

**Model response**

Answer: From the premises, we can conclude that moon 2 is visible to the naked eye.

**Parsed model response**

$\therefore \{\mathsf{visibleToTheNakedEye}(moon2)\}$

---

The repetition of "moon 2 is visible to the naked eye" in the second disjunct causes claude-3.7-sonnet to suppress a consistent alternative (that asteroid B is visible to the naked eye and moon 2 is not), leading to a fallacious conclusion.

**Example 2: Invalid existential instantiation (`microsoft/phi-3-mini-128k-instruct`)**

**Question**

I'm working in a materials science lab and we've gotten some puzzling results. I need to use logical reasoning to figure out what's going on. Here's what I know so far:

- Either darkonium is not radioactive, or darkonium is radioactive.
- Either voidite is electrically insulating, or voidite is not self-repairing and darkonium is not radioactive.
- There is some X such that X is self-repairing.
- Voidite is self-repairing and voidite is not radioactive.
- Either voidite is not electrically insulating, or voidite is corrosive and voidite is electrically insulating.

For the purpose of this question, I want you to write what follows in English. Please be succinct, precise and clear in your answer. Write a logical statement of the form "Answer: From the premises, we can conclude that . . . " and then clearly write your conclusion. Please be succinct, precise, and clear.
What if anything follows? I do not have an intended answer in mind, and it is possible that nothing follows. Please be succinct and precise.
I want you to answer immediately. Read the question and provide your answer in the format given.
What follows? Answer in the format that I showed you. Write "Answer: {logical statement}".

**ETR representation**

- $\{\overline{\text{radioactive}(darkonium)}, \text{radioactive}(darkonium)\}$
- $\{\text{electricallyInsulating}(voidite), \overline{\text{radioactive}(darkonium)}\text{selfRepairing}(voidite)\}$
- $\exists x\{\text{selfRepairing}(x)\}$
- $\{\text{selfRepairing}(voidite)\overline{\text{radioactive}(voidite)}\}$
- $\{\overline{\text{electricallyInsulating}(voidite)}, \text{corrosive}(voidite)\text{electricallyInsulating}(voidite)\}$

**ETR predicted fallacious conclusion**

$\therefore \{\overline{\text{radioactive}(darkonium)}\text{electricallyInsulating}(voidite)\text{corrosive}(voidite)\}$

> **Model response**
>
> Answer: From the premises, we can conclude that darkonium is not radioactive.
> Explanation:
>
> 1. From the third premise, we know there exists some X that is self-repairing.
> 2. From the fourth premise, we know voidite is self-repairing.
> 3. Therefore, voidite is the X mentioned in the third premise.
> 4. From the second premise, since voidite is self-repairing, it must be true that voidite is not self-repairing and darkonium is not radioactive.
> 5. However, we know from the fourth premise that voidite is self-repairing, which contradicts the second premise.
> 6. Therefore, the only way to resolve this contradiction is if the second premise is false, meaning that voidite is self-repairing and darkonium is not radioactive.
> 7. From the first premise, we know that either darkonium is not radioactive, or it is radioactive.
> 8. Since we have established that darkonium is not radioactive, the first premise is satisfied.
> 9. Therefore, we can conclude that darkonium is not radioactive.
>
> **Parsed model response**
>
> $\therefore \{\overline{\mathsf{radioactive}(darkonium)}\}$

phi-3-mini-128k-instruct's third conclusion ("Therefore, voidite is the X mentioned in the third premise.") is not logically valid. This conclusion forces the LLM to fallaciously endorse a contradiction (that voidite both is and is not self-repairing), after which the LLM decides to ignore the second premise and conclude a fallacy.

# E    EXAMPLE OF FALLACY BLOCKING THROUGH PREMISE ORDER REVERSAL

> **Original premise order (`google/gemma-2-9b-it`)**
>
> Question
>
> I'm a researcher studying newly discovered psychic abilities. I need to understand their interactions through logical reasoning. Here's what we know:
>
> - Either reality warping is matter-moving, or precognition is space-bending.
> - Either reality warping is matter-moving, or reality warping is not matter-moving.
>
> For the purpose of this question, I want you to write what follows in English. Please be succinct, precise and clear in your answer. Write a logical statement of the form "Answer: From the premises, we can conclude that . . ." and then clearly write your conclusion. Please be succinct, precise, and clear.
> What if anything follows? I do not have an intended answer in mind, and it is possible that nothing follows. Please be succinct and precise.
> I want you to answer immediately. Read the question and provide your answer in the format given.
> What follows? Answer in the format that I showed you. Write "Answer: {logical statement}".

**ETR representation**

- $\{\mathsf{matterMoving}(reality\,Warping), \mathsf{spaceBending}(precognition)\}$
- $\{\mathsf{matterMoving}(reality\,Warping), \overline{\mathsf{matterMoving}(reality\,Warping)}\}$

**ETR predicted fallacious conclusion**

$\therefore \{\mathsf{matterMoving}(reality\,Warping)\}$

**Model response**

Answer: From the premises, we can conclude that reality warping is matter-moving.

**Parsed model response**

$\therefore \{\mathsf{matterMoving}(reality\,Warping)\}$

In the original premise order, gemma-2-9b-it falls for a standard ETR-predicted fallacy by fixating on the repeated predicate in the second premise.

---

**Reversed premise order (`google/gemma-2-9b-it`)**

**Question**

I'm a synthetic biology researcher studying advanced bioengineered life forms. I need to understand their capabilities through logical analysis. Here's what we've created:

- Either nanohive is swarm-forming, or nanohive is not swarm-forming.
- Either nanohive is swarm-forming, or chronoplast is quantum-computing.

For the purpose of this question, I want you to write what follows in English. Please be succinct, precise and clear in your answer. Write a logical statement of the form "Answer: From the premises, we can conclude that . . ." and then clearly write your conclusion. Please be succinct, precise, and clear.
What if anything follows? I do not have an intended answer in mind, and it is possible that nothing follows. Please be succinct and precise.
I want you to answer immediately. Read the question and provide your answer in the format given.
What follows? Answer in the format that I showed you. Write "Answer: {logical statement}".

**ETR representation**

- $\{\mathsf{swarmForming}(nanohive), \overline{\mathsf{swarmForming}(nanohive)}\}$
- $\{\mathsf{swarmForming}(nanohive), \mathsf{quantumComputing}(chronoplast)\}$

**ETR predicted fallacious conclusion**

$\therefore \{\mathsf{swarmForming}(nanohive)\}$

**Model response**

Answer: From the premises, we cannot definitively conclude anything about whether nanohive is swarm-forming or not.

**Parsed model response**

$\therefore \{\emptyset\}$

The **Reversed premise order** example displays the exact same logical problem (with mapping to a different natural language theme) except the premise order is flipped. In this instance, gemma-2-9b-it is able to avoid the fallacy it made earlier and not conclude anything.

