# OpenReview forum: "Theory-Grounded Evaluation of Human-Like Fallacy Patterns in LLM Reasoning"
_ICLR.cc/2026/Conference — ICLR 2026 Poster_

### Official Review · Reviewer_k8G8 · 2025-10-29

**Soundness:** 1
**Presentation:** 1
**Contribution:** 2
**Rating:** 0
**Confidence:** 4

**Summary:**

### Summary: *Analyzing LLMs through the Erotetic Theory of Reasoning (ETR)*

The paper analyzes large language models (LLMs) through the lens of the **Erotetic Theory of Reasoning (ETR)** — a framework developed by philosophers and logicians to study how questions relate to one another.

### **Background: Key Idea of ETR**
ETR focuses on how the **answer to one question can resolve or transform another question**.
For example:

> “If the light is on, then the power is on.”

This statement evokes two questions:
1. Is the light on?
2. Is the power on?

Answering *yes* to the first question naturally resolves the second one, concluding that the power is on.

### **Experimental Setup**
The authors use **PyETR**, a tool that automatically derives ETR structures from statements, to generate prompts composed of:
**theme introduction**, a set of **premises**, and a follow-up **question** asking what logically follows from those premises.
They source their ETR style reasoning problems from Reason and Inquiry text (Koralus, 2022).

### **Evaluating Logical Correctness of Model Response**
To test logical correctness, they use **PySMT**, a logic solver already open-sourced for the community.
The paper measures the logical correctness by setting the criterion that if the **negation** of a model’s conclusion contradicts the premises, the conclusion is considered **logically correct**.

### **Evaluating Human-like Fallacies**
To assess human-like reasoning errors, the authors check whether the model’s answer matches ETR’s prediction of what a human would infer,  but is not logically correct. These cases are labeled as ETR-predicted fallacies.
Finally, the **fallacy rate** is defined as the proportion of these human-like fallacies among all logically incorrect responses.

### **Key Findings of This Paper**
- There is a **negative correlation** between logical ability and fallacy rate:
  **Stronger models are more likely to make human-like reasoning mistakes** when they do err.
- Reversing the **order of premises** in the same logical problem often **reduces fallacy production**, showing that LLMs exhibit **order effects** similar to humans.

**Strengths:**

This paper draws ETR reasoning theory into evaluating LLM behavior, suggesting a viewpoint that logically strong models in terms of reasoning ability are more prone to making human-like fallacy errors. This is evaluated by employing existing open-source implementation of ETR logic solver, PyETR, and existing dataset "Reason and Inquiry text".

**Weaknesses:**

### **1. Low-Quality Prompt Design**

The prompts generated for evaluation appear to be poorly constructed, making it unclear whether models could interpret or respond to them properly.

**Evidence**: The reported correlation between model performance (as measured by LLM Elo scores) and logical correctness is nearly zero
(r = 0.004, p = 0.981; ρ = −0.04, p = 0.777). This indicates that the dataset may not reliably capture reasoning ability, suggesting that prompt design, rather than model reasoning, could be the limiting factor.
This undermines the paper’s main claim about fallacy trends across model capabilities, since a dataset that fails to differentiate between weak and strong models cannot support such conclusions.

### **2. Lack of Methodological Clarity**

Several crucial details are omitted or insufficiently explained, making it difficult to assess or reproduce the study.

a. The statement “Pre-analysis integrity checks identified 17 items whose fallacy status did not meet the final specification” is ambiguous. The paper never defines what “final specification” refers to or how it was determined.

b. The authors mention creating “12 themes for natural language framings of logical problems,” yet these themes are not listed, exemplified, or referenced in the appendix or main text. Such omissions significantly reduce transparency and reproducibility.

c. The criterion for logical correctness (checking whether the negation of the conclusion contradicts the premises) is introduced without any justification or explanation of the underlying logic principle. While the negation-contradiction test is a valid formal method for entailment checking, it should have been theoretically motivated and explained for readers unfamiliar with formal logic or SMT solvers.

### **3. Conceptual and Logical Weaknesses**

The paper’s contribution is mainly an application of an existing psychological framework (Erotetic Theory of Reasoning, ETR) to LLM outputs, with little novel insight into model-specific reasoning mechanisms. It feels more like a reinterpretation of known model behaviors through an old theoretical lens rather than a discovery of new phenomena.

There is a logical circularity in the argument: the paper assumes that ETR is a valid description of human reasoning and then uses it as both the theoretical predictor and evaluation criterion for LLMs. This conflates explanatory alignment with empirical validation — the models might appear “ETR-like” simply because the evaluation framework enforces it.

The study treats fallacy alignment with ETR as evidence of human-likeness, but it never justifies why human-like fallacies should increase with logical ability. This causal interpretation is speculative and unsupported by formal or cognitive reasoning theory.

### **4. Poor Exposition and Notation Clarity**

Section 2, which introduces ETR, is very difficult to follow. The text uses non-standard and undefined notations without clear definitions or examples. The presentation of PyETR is redundant and unfocused, repeating information available in prior works without connecting it directly to this study’s experimental design.

Moreover, the citation style is inconsistent: citations should be enclosed in parentheses unless grammatically integrated into the sentence. Several sentences lack periods or contain minor grammatical errors, making the text unnecessarily difficult to read. These stylistic and formatting lapses reflect insufficient editorial care, which reduces the paper’s overall readability and professionalism.

### **5. Lack of rigorous experimental design and analysis**
There is no ablation or control analysis demonstrating that the observed “fallacy-blocking” from reversing premises is statistically robust or independent of surface-level linguistic cues. The paper conflates logical validity (a formal property) with psychological plausibility (a behavioral trait), without acknowledging that these are distinct dimensions of reasoning.

**Questions:**

The questions mainly concern the clarity of the notations, the ETR-style logical expressions, and the undefined logical operators that appear throughout the paper.
For example, what does “p” represent in the reported Pearson/Spearman correlations? I assume it refers to the p-value, but such notation should be explicitly defined, not used without explanation.

---

> ### Author Response · Authors · 2025-11-27
> **Response for Reviewer k8g8 [1/2]**
>
> Thank you for the detailed review. There are several apparently serious criticisms, but we believe these stem from a misunderstanding of the genre of our work and we are happy to clarify.
>
> To begin with, the example of lights and power is not a good representation of the key ideas of ETR. ETR does indeed focus on "how the answer to one question can resolve or transform another question," but this is a feature of deductive systems in general. ETR states that human reasoners treat incoming premises as answers to implicit questions _as directly as possible_ (hence the emphasis on filtering alternatives and the resultant fallacious conclusions). We refer to the example in section 2 of our paper.
>
> This misunderstanding is not crucial, though, as our interest in ETR for this paper is as a generative proxy for human-like reasoning, _not_ as a unique mechanistic explanation of the human reasoning process. That generative proxy is operationalized by PyETR, effectively black-boxing the theoretical detail.
>
> To list the serious concerns that we should address:
> - There is the complaint that no light is shed on model-specific reasoning mechanisms, along with perhaps an expectation that the importance of ETR is to provide some mathematical skeleton for that mechanistic explanation.
> - There is the suggestion of a logical circularity, in that we are assuming the goodness of ETR as a descriptive theory and then using it with explanatory evaluative intent on model.
> - We are further charged with making speculative and unsupported causal claims that human-likeness inevitably increases alongside logical ability.
>
> We will now try to restate our core methodology and claims in a way that hopefully addresses these concerns.
>
> We know for an empirical fact that when humans make logical reasoning errors, those errors are not random, but structured. These structured errors are called fallacies. In our work, we question whether LLMs also commit such fallacies. Either case would be interesting: if they do not, then that is evidence LLMs are behaviourally quite different from human reasoners, and if they do, then there is a point of behavioural similarity (regardless of what dissimilarities occur mechanistically). So, we want to measure this human-like-fallacy-proneness across many models. ETR is a formal theory, operationalized as code in PyETR, which conservatively generalizes a very wide basis of empirical observation of this human-like-fallacy-proneness (the proof of this empirical alignment is shown throughout Koralus (2022) and prior work). PyETR allows us to answer our question systematically. What we find is that as models get "stronger" (proxied by Chatbot Arena ELO, because model internals can all be very different and there is no apples-to-apples comparison), models indeed make more human-like-fallacies.
>
> More specifically, we thank you for several helpful critiques and suggestions, which we will address in the paper as outlined below:
>
> ## 1. Dataset does not discriminate between weak and strong models
> Logical correctness on this dataset is not expected to correlate with Arena Elo, because the dataset is deliberately constructed so that every item has an ETR-predicted fallacy. The finding is that, conditional on being logically wrong, stronger models manifest the specific fallacies predicted by ETR at a higher rate. We can make this implication of our dataset construction clearer in several places in the text.
>
> ## 2. Poor prompt design explains the results
> Our intervention study on premise order shows this to be untrue. Reversing premise order produced statistically significant fallacy-blocking across most models, which only occurs if prompts are being processed meaningfully. All prompts follow a tight template, and evaluation is done by translating model outputs back into ETR views and validating with an SMT solver. Across 38 models the parse-error rate was far below threshold—we only omitted one model due to parsing issues—indicating the models could interpret and respond to the prompts.
>
> ## 3. “Final specification” and curation concerns
> The “final specification” refers to criteria defined in section 3.2: (1) single-alternative ETR conclusion, (2) conclusion corresponds to a fallacy under PyETR’s default procedure, and (3) total atoms within the 4–11 bound. The 17 excluded items failed one of these criteria. We will add this explicit cross-reference.
>
> ## 4. Missing list of 12 themes
> The themes were omitted due to space, not withheld. We will include the full list and mapping table in the appendix. Theme choice does not affect the formal structure: all PyETR views are isomorphic across themes, which was the point of including them as contamination-resistance, not as experimental degrees of freedom.

---

> > ### Author Response · Authors · 2025-11-27
> > **Response for Reviewer k8g8 [2/2]**
> >
> > ## 5. Logical correctness criterion unjustified
> > Checking entailment by testing whether (premises ∧ ¬conclusion) is unsatisfiable is very standard (PySMT solves exactly this). We will add a sentence clarifying this.
> >
> > ## 6. Circularity in using ETR predictions
> > Again, to the points made above, our study does not assume ETR is the correct theory of human reasoning. ETR is used only as a generator of fallacy-eliciting items and as a labeling oracle for predicted fallacies. The empirical claim is conditional: “given ETR’s predictions, do stronger models disproportionately match them?” No assertion is made about the mechanistic truth of ETR for humans or models. The goodness of ETR as a theory of human reasoning we consider to be argued and settled in the literature outside the scope of this paper; it would be unreasonable to expect every paper to be a standalone first-principles relitigation.
> >
> > ## 7. Why should human-like fallacies increase with capability?
> > We do not claim a causal theory; we report the correlation. Multiple interpretations are explicitly acknowledged in section 5. The paper does not argue that this is normatively desirable or that scaling causes fallacy-likeness; only that error composition shifts reliably.
> >
> > ## 8. Notation/exposition deficiencies
> > The ETR definitions are reproduced verbatim (with permission) because they are the canonical version. We will refactor section 2 to clarify the minimum concepts necessary for this study. Citations and minor grammatical issues will be corrected.

---

### Official Review · Reviewer_mN8S · 2025-11-01

**Soundness:** 3
**Presentation:** 3
**Contribution:** 3
**Rating:** 8
**Confidence:** 3

**Summary:**

Note that I have previously reviewed this paper for another conference. The paper is substantially similar, and I copy some of the review here.

This paper shows that language models which are “stronger” (according to Chatbot Arena Elo score) produce more “human-like” errors, specifically those described by the “erotetic theory of reasoning” (ETR), which posits that humans reason by maintaining disjunctive alternatives and filtering these alternatives when taking on new information.

**Strengths:**

I find the results and analyses to be sound, interesting, and worth sharing.
* interesting and important result about how models interpret disjunctive normal forms in ways similar to humans
* the use of the ETR model is novel and the analyses are sound

I also appreciate that the authors have addressed my previous concerns, copied here:
* The Chatbot Arena may not be the best proxy for "strong" models, given that human preferences on the arena may be prone to surface features like formatting (which the arena tries to account for) and sycophancy. Recent work has also demonstrated the fallibility of the leaderboard (https://arxiv.org/abs/2504.20879). I think a plausible alternate interpretation of the results is that Chatbot Arena measures (amoong other things) how well a model interprets natural language in the way that humans do. Given that humans interpret disjunctive normal forms in a particular way, the ELO score reflects this. Thus, intead of speaking to patterns in the behavior of models of various "strength", these results might instead be speaking more to what Chatbot Arena Elo is measuring
* I think it’s a bit misleading to call it an inverse scaling law, which most people would I think interpret as indicate worse absolute performance with scaling. Rather, here the result is about the proportion of errors that fit the etoteric pattern
* I may have misunderstood the methods, but I believe that the evaluation set mostly revolves around disjunctive normal forms and slight variations thereof. I think this is a relatively narrow set of behaviors, and the claim in the title may be overreaching given this narrowness

**Weaknesses:**

No substantial concerns

**Questions:**

I would also be interested to see the analysis separately for non-reasoning and “reasoning” models (esp those models trained via long-range RL, such as the o-series from OpenAI). And also models that are trained more heavily on e.g. code rather than human language. But this is not so much a concern as a curiosity.

---

> ### Author Response · Authors · 2025-11-27
> **Response for Reviewer mN8S**
>
> We appreciate your clear positive evaluation.
>
> ## 1. On Chatbot Arena and interpretation
> We agree that Elo may partly reflect human-alignment in language interpretation. This is why section 5 now stresses that the results do not imply a mechanistic scaling law; they reflect an empirical regularity under capability proxies used in current practice.
>
> ## 2. On the scope of logical forms
> The dataset centers on monadic DNF-style views to ensure PyETR tractability and to minimize confounds from mapping polyadic views to natural language. We agree this is a narrow but controlled slice, and we will consider a title and abstract adjustment to reflect the scope more precisely.

---

### Official Review · Reviewer_EveU · 2025-11-02

**Soundness:** 2
**Presentation:** 2
**Contribution:** 2
**Rating:** 4
**Confidence:** 4

**Summary:**

This paper uses the ETR to evaluate whether LLM errors match human fallacy patterns across 383 reasoning problems and 38 models. The key finding is,  higher-capability models (Chatbot Arena Elo score) produce a significantly larger proportion of human-like fallacies (ρ = 0.360, p = 0.0265),  but no improvement in overall correctness. This suggests scaling increases alignment with human error patterns rather than improving logical reasoning accuracy.

**Strengths:**

1. This paper introduces a mature method for evaluating human fallacies, which has a solid theoretical foundation and provides good reproducibility based on the open-source pyETR implementation.
2. Extensive experimental validation was conducted on a wide range of models (of different sizes and architectures), and the results are quite comprehensive.
3. Insightful experimental designs explored how human thought patterns manifest in large-scale models.

**Weaknesses:**

1. Experiments indicate that the moderately low correlation weakens the experimental claims put forward in the paper.
2. The paper lacks theoretical and principled analysis, focusing instead on assessing the similarity between LLMs and humans in reasoning errors and presenting observed experimental results. However, it fails to explore potential causes for this phenomenon or propose solutions to the problem, resulting in limited practical applicability.

**Questions:**

1. This paper evaluates model performance solely through the relatively comprehensive Elo metric. However, it remains unclear whether the reasoning fallacy test is more inclined toward models capable of generating long reasoning chains, or whether it is more closely related to the model's ability to answer questions in the STEM domain.
2. Could a more comprehensive set of metrics be employed to holistically evaluate the model's performance, encompassing aspects such as answer diversity and reasoning capabilities? Further analysis of the relationship between human-like errors and these capabilities would strengthen the article's conclusions.

---

> ### Author Response · Authors · 2025-11-27
> **Response for Reviewer EveU**
>
> Thank you for your constructive assessment.
>
> ## 1. Correlation is moderate?
> The effect size is moderate because error composition is inherently noisy across heterogeneous model families. This is in contrast to other correlational claims about LLM performance, like traditional scaling laws [1], where homogeneous architectures exhibit "cleaner" effects. The point of our study is not to show such large-magnitude effects but a statistically robust consistency across architectures. This is precisely what we observe (ρ = 0.360, p = 0.0265 across 38 diverse models), and we explicitly avoid any strong causal interpretation.
>
> We state this caveat in section 5:
> > Though the correlation strength (ρ = 0.360) may appear moderate, it is robust across diverse model architectures and training paradigms, suggesting a fundamental relationship rather than artifact.
>
> ## 2. Lacks theoretical explanation?
> The goal is empirical characterization. We avoid mechanistic claims intentionally because multiple plausible explanations (summarized in section 5) remain open: training-data exposure, convergent pruning of alternatives, RLHF-driven conversational consistency, etc. We appreciate this point, and will make this positioning more explicit in section 5.
>
> ## 3. Only Elo used as capability metric?
> We actually test the result against three capability proxies (as stated in section 1):
> 1. Chatbot Arena Elo,
> 2. HELM capabilities composite,
> 3. EpochAI training-compute estimates.
> The correlation holds for all available metrics. We will make these results more prominent in the results in section 4.
>
> ## 4. Would richer metrics help?
> Yes. Our dataset is constructed so that correctness is not the main discriminant variable; fallacy-composition is. We will expand section 5 to highlight future evaluation axes: chain-length sensitivity, answer diversity, model-specific reasoning profiles, etc.
>
> [1] Kaplan, J., McCandlish, S., Henighan, T., Brown, T. B., Chess, B., Child, R., Gray, S., Radford, A., Wu, J., & Amodei, D. (2020). Scaling Laws for Neural Language Models. arXiv. https://arxiv.org/abs/2001.08361.

---

### Official Review · Reviewer_J4ty · 2025-11-03

**Soundness:** 2
**Presentation:** 3
**Contribution:** 2
**Rating:** 6
**Confidence:** 2

**Summary:**

The paper looks into whether LLMs generate predictable fallacies like humans using ETR. Using the PyETR framework, authors generate synthetic reasoning problems and evaluate 38 LLMs. Their findings indicate that as models get more capable (with chatbotarena elo as a proxy) the predictability of the inaccurate responses increases (they get more human-like). The paper highlights that more capable models don’t just make fewer errors, they make more human-like ones.

**Strengths:**

1.  I resonate well with evaluating the human-likeness of reasoning of LLMs - LLMs are trained with trillions of tokens that are human written, and it is expected that human cognitive biases should show in their generations. And in that sence I find the paper quite interesting

2. The paper also brings in the research question of predictability of errors. And shows that more capable model are more predictable as well

3. Authors have done a good job at addressing a bigger audience than the ones familiar with cog-sci. I feel the paper is written well enough for someone not familiar with ETR to read and appreciate this. And I commend this.

That being said, I have concerns too. Please see weaknesses.

**Weaknesses:**

Weakness:

1. My first and major concern is the scope of the work. I think the section 4 needs more insights. I feel the scope is too limited in its current form.

2. I think the paper can benefit from some anecdotal analysis, i.e. if the authors could give more insights of the form "the reasoning pattern in model X is like Y, and hence …." instead of a high level correlation metric reported.

**Questions:**

Please see weakness

---

> ### Author Response · Authors · 2025-11-27
> **Response for Reviewer J4ty**
>
> Thank you for the constructive assessment.
>
> ## 1. Scope and depth of section 4
> We clarify the scope of the empirical contribution by foregrounding the two analyses already present: the fallacy–capability correlation and the intervention study on premise-order reversal (p. 8). The dataset includes multiple families of fallacies (e.g., suppressed-alternative collapse [1], quantifier mis-scoping [2]), and to make this structure more visible we will add a brief set of illustrative examples that show how different models handle these representative items. This provides more texture to the aggregate results without changing the core methodology.
>
> ## 2. Anecdotal/interpretive examples
> To aid interpretability, we will include a small appendix box with a few short model traces:
> - a case where weaker models give unrelated outputs,
> - a case where stronger models follow the ETR-predicted suppression path,
> - a case where premise-order reversal blocks the fallacy.
>
> These lightweight examples make the reasoning patterns easier to inspect without altering the overall analysis.
>
> [1] Koralus, P. (2022). Reason and Inquiry: The Erotetic Theory (1st ed.). Oxford University Press. https://doi.org/10.1093/oso/9780198823766.001.0001. p. 61.
>
> [2] Ibid. p. 131.

---

### Author Response · Authors · 2025-12-03
**Summary of revisions**

Dear Area Chairs,

Thank you for your attention to our work. As the reviewer discussion period has been cut short, we will summarise the feedback we have received and the changes we have implemented in response. We hope that this will aid in your decision process.

## J4ty
Reviewer J4ty found the paper conceptually well-motivated, addressing an important topic in the human-likeness of LLM reasoning, and accessible to general AI/ML readership outside of cognitive science. Their review critiqued our results section for being too limited in scope and lacking anecdotal analysis.

## EveU
Reviewer EveU found our methodological approach to be solid, reproducible, and extensively validated. They also commended the insightful findings about human-like LLM reasoning. They critiqued the low magnitude of our correlation result, as well as our avoiding any causal explanation for our results.

## mN8S
Reviewer mN8S found our results important and interesting, and our use of the ETR model to be both novel and methodologically sound. They did not list any substantial concerns.

## k8G8
Reviewer k8G8 raised significant criticisms about our basic conceptual design, overall methodology, interpretation of results, exposition, and notational clarity.

# Revisions

In response to our reviews, we have made revisions to the presentation and discussion of our results, as well as substantial additions to our appendices. We outline these changes below:

1. Reviewer J4ty requested anecdotal error analysis to aid the interpretation of our results. We add **Appendix D**, which provides examples of different fallacy types committed by LLMs on our dataset, and **Appendix E**, which presents a concrete example from our premise order reversal experiment where a fallacy is blocked.
2. Reviewer J4ty also requested improvements to the scope of section 4. We revised the language in this section for clarity and add a mention to a new **Appendix C**, which provides a comparison to other capability proxies (composite HELM scores and EpochAI training compute estimates).
3. Reviewer EveU requested a more comprehensive set of metrics in our evaluation. We again point to the new **Appendix C**, which augments our central claim, and expand section 5 to discuss additional metrics that could be considered in future work.
4. Reviewer EveU also requested more theoretical and principled analysis. We revised the language in section 5 for clarity, specifically highlighting that our results provide no mechanistic explanation but point to several that could be considered in future work.
5. Reviewer k8g8 raised helpful feedback about inconsistent citation style and grammatical errors. We amend these in several places throughout the work.
6. Reviewer k8g8 also requested transparency into our 12 themes for natural language framings. We add these in a new **Appendix B**, which contains each complete ontology and discusses the choice of objects and predicates to ensure that logical consistency is preserved in the natural language.
7. Reviewer k8g8 also suggested several places where signposting in the text could be clearer. We make small amendments through sections 2 and 3 to improve this aspect.

We regret that we were unable to discuss these changes further with our reviewers. We feel confident that we have thoroughly addressed the critiques raised by reviewers J4ty and EveU, and would have desired to see the affected changes in their final scores. Ultimately we are grateful for their feedback as we feel we have arrived at a substantially stronger and more thorough presentation of our work. Thank you for your attention to our paper.

---

### Meta-Review · Area_Chair_kbi8 · 2025-12-27

**Summary:**

The paper investigates whether LLMs generate predictable fallacies, as humans do, using ETR. Reviewers agree that this paper proposes a novel perspective on evaluating a model's behavior. I have carefully read all discussions, especially the authors' response to reviewer k8G8, and believe that, although the submitted manuscript lacks some details, it makes a unique contribution to the community. In the revision, the authors have already added more details to the appendix. Hence, I would recommend accepting this paper, but I am comfortable with its rejection.

**Reviewer Concerns:**

The primary concern from reviewers is the evaluation design (e.g., prompts, baseline models)

**Reviewer Scores:**

Based on the rebuttal, I think reviewer k8G8 could improve the overall score a bit (from 0 to 2/4), but probably not enough to a positive score (6).
For other reviewers, I think they will maintain their original score.

---

### Decision · Program_Chairs · 2026-01-26

Accept (Poster)